# RESEARCHRUBRICS: A BENCHMARK OF PROMPTS AND RUBRICS FOR EVALUATING DEEP RESEARCH AGENTS

**Manasi Sharma** [*,1]**, Chen Bo Calvin Zhang** [1]**, Chaithanya Bandi** [1]**, Clinton Wang** [†],
**Ankit Aich** [1]**, Huy Nghiem** [2]**, Tahseen Rabbani** [3]**, Ye Htet** [4]**, Brian Jang** [1]**, Sumana Basu** [5]**,
**Aishwarya Balwani** [1]**, Denis Peskoff** [6]**, Marcos Ayestaran** [1]**, Sean M. Hendryx** [†]**,
**Brad Kenstler** [1]**, Bing Liu** [1]

[1]Scale AI    [2]University of Maryland    [3]University of Chicago
[4]Washington University, St. Louis    [5]McGill University    [6]University of California, Berkeley
[†]Work conducted while at Scale AI

## ABSTRACT

Deep Research (DR) is an emerging agent application that leverages large language models (LLMs) to address open-ended queries. It requires the integration of several capabilities, including multi-step reasoning, cross-document synthesis, and the generation of evidence-backed, long-form answers. Evaluating DR remains challenging because responses are lengthy and diverse, admit many valid solutions, and often depend on dynamic information sources. We introduce RESEARCHRUBRICS, a standardized benchmark for DR built with over 2,800+ hours of human labor that pairs realistic, domain-diverse prompts with 2,500+ expert-written, fine-grained rubrics to assess factual grounding, reasoning soundness, and clarity. We also propose a new complexity framework for categorizing DR tasks along three axes: conceptual breadth, logical nesting, and exploration. In addition, we develop human and model-based evaluation protocols that measure rubric adherence for DR agents. We evaluate several state-of-the-art DR systems and find that even leading agents like Gemini's DR and OpenAI's DR achieve under 68% average compliance with our rubrics, primarily due to missed implicit context and inadequate reasoning about retrieved information. Our results highlight the need for robust, scalable assessment of deep research capabilities, to which end we release RESEARCHRUBRICS (including all prompts, rubrics, and evaluation code) to facilitate progress toward well-justified research assistants.

🤗 https://huggingface.co/datasets/ScaleAI/researchrubrics

## 1 INTRODUCTION

An exciting development in the growing capabilities of large language models (LLMs) is the emergence of Deep Research agents: autonomous LLM-based systems that conduct multi-step web exploration, targeted retrieval, and synthesis to answer open-ended queries. Industry leaders have begun deploying such systems (e.g., OpenAI's "Deep Research" OpenAI (2025a) and Google's "Gemini Deep Research" Google (2025)), which have demonstrated strong performance on certain benchmarks (for instance, scoring 26.6% on the expert-level HLE benchmark Phan et al. (2025)). However, evaluating deep research agents poses significant challenges. Deep Research (DR) tasks are inherently open-ended: they require reasoning across multiple documents, often with no single "correct" answer, and their outputs can be long and varied. Consequently, existing evaluation methods fall short. Typical QA benchmarks, both general Yang et al. (2018); Mialon et al. (2023); Phan et al. (2025); Krishna et al. (2025) and deep research specific Java et al. (2025); Coelho et al. (2025), focus on short, easily-verifiable factual answers and do not capture the long-form, multi-source synthesis

---

[*]Correspondence to: manasi.sharma@scale.com.

required by DR, e.g., *"Which material has band gap* $0.9\,\text{eV}$*, dislocation density* $4 \times 10^8 \text{cm}^{-2}$*?"* with the unique answer *"Gallium nitride (GaN)"*.

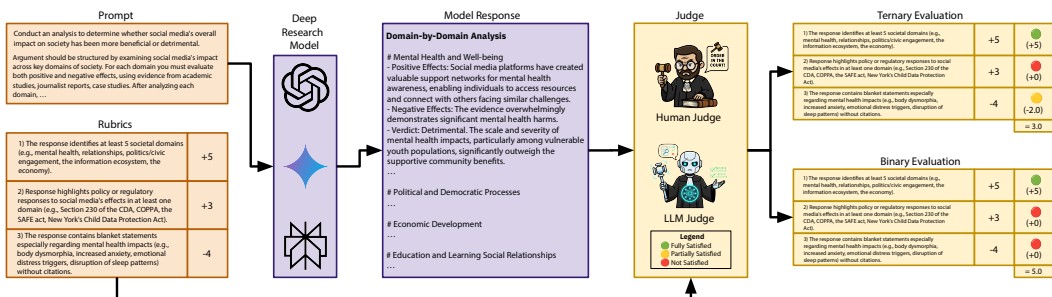

Figure 1: Overview of RESEARCHRUBRICS and its evaluation pipeline.

Several recent efforts to benchmark deep research agents directly have also revealed important limitations: for example, some benchmarks introduce LLM-generated rubrics and evaluation metrics reliant upon LLM-generated reference reports Du et al. (2025), thus raising concerns about circularity and limited oversight Dorner et al. (2025), while others are far more narrow in their scope, assessing only one specific angle of research in a technical domain (e.g., generating a "Related Works" section) Patel et al. (2025); Li et al. (2025); Wan et al. (2025). In practice, however, users direct deep research systems toward a broad array of everyday topics, ranging from business reports to consumer-related queries, underscoring the need for benchmarks that combine domain diversity with expert-authored, fine-grained rubrics.

To better characterize these challenges and motivate our approach, we introduce a **task complexity framework** for deep research. Each query can be described along three independent axes: (1) its **conceptual breadth** (the number and diversity of distinct topics or domains involved), (2) its **logical nesting depth** (the number of reasoning or decision steps required, including sub-questions and conditionals), and (3) its **exploration level** (the degree of open-endedness or underspecification of goals). This tri-axial view highlights how DR queries differ from simpler QA tasks and helps articulate the shortcomings of existing methods: simple QA benchmarks lack sufficient breadth, depth, and exploration, while many current DR benchmarks fail to cover this full, multi-axial complexity.

We introduce RESEARCHRUBRICS, which pairs realistic, diverse prompts with expert-authored, fine-grained rubrics for deep research. We curate queries from nine broad domains (including business planning, historical analysis, technical documentation, and common consumer questions) to reflect real-world use cases. Each prompt comes with a detailed rubric: in total, we provide 2,593 rubric criteria that check factual grounding, coherence of reasoning, completeness, relevance, and clarity of the answer. The benchmarks also include negative rubrics that specifically aim to penalize extraneous or incorrect content. Importantly, all rubrics are written and reviewed by human experts (not auto-generated), ensuring they capture nuanced, domain-specific requirements. We also develop evaluation protocols for both human and automated scoring. Following the LLM-as-judge paradigm, we use powerful LLMs to assess rubric compliance, and we systematically experiment with improving this process comparing binary vs. ternary grading for each criterion and the level of detail in the rubrics. Finally, we apply our framework to leading DR systems (OpenAI's DeepResearch OpenAI (2025a), Google Gemini's Deep Research Google (2025), and Perplexity's Deep Research AI (2025)). The results show that even the strongest agents fall below 68% average rubric compliance, revealing substantial room for improvement in multi-document synthesis and rigorous justification.

**Our contributions** are as follows:

- **A human-crafted benchmark for deep research.** We present RESEARCHRUBRICS, a suite of open-ended research tasks across diverse domains, each with an expert-written rubric (2,593 total criteria). Crucially, each rubric is both written and reviewed by humans, thereby mitigating potential anchoring biases that may arise when only verifying LLM-generated rubrics.

- **A task complexity framework.** We formalize deep research queries along three axes—**breadth**, **depth**, and **ambiguity**—to distinguish them from conventional QA tasks and to guide the construction of balanced benchmarks that reflect real-world deep research queries.

Table 1: Comparison of RESEARCHRUBRICS with representative Deep Research benchmarks.

| Benchmark | Human-authored Rubrics | Expert-Curated | Open-Ended Tasks | Non-Technical Domains | LLM-as-Judge | Average # Rubrics per task |
|---|---|---|---|---|---|---|
| AcademicBrowse Zhou et al. (2025) | ✗ | ✗ | ✗ | ✓ | ✗ | – |
| BrowseComp Wei et al. (2025) | ✗ | ✗ | ✗ | ✓ | ✗ | – |
| ResearchBench Liu et al. (2025) | ✗ | ✗ | ✗ | ✓ | ✗ | – |
| ResearcherBench Xu et al. (2025) | ✓ | ✓ | ✓ | ✗ | ✓ | 14 |
| DeepScholar-Bench Patel et al. (2025) | ✗ | ✗ | ✓ | ✗ | ✓ | – |
| ReportBench Li et al. (2025) | ✗ | ✗ | ✗ | ✓ | ✓ | – |
| DeepResearch Bench Du et al. (2025) | ✗ | ✓ | ✓ | ✗ | ✓ | 25 |
| Mind2Web2 Gou et al. (2025) | ✗ | ✓ | ✗ | ✓ | ✓ | 50 |
| LiveResearchBench Wang et al. (2025) | ✗ | ✓ | ✓ | ✓ | ✓ | – |
| LiveDRBench Java et al. (2025) | ✗ | ✗ | ✗ | ✓ | ✓ | – |
| ExpertLongBench Ruan et al. (2025) | ✓ | ✓ | ✓ | ✓ | ✓ | 16 |
| DeepResearch Arena Wan et al. (2025) | ✗ | ✗ | ✓ | ✓ | ✓ | – |
| DeepResearchGym Coelho et al. (2025) | ✗ | ✗ | ✓ | ✓ | ✓ | – |
| SPOT Son et al. (2025) | ✓ | ✗ | ✗ | ✗ | ✓ | – |
| RESEARCHRUBRICS (Ours) | ✓ | ✓ | ✓ | ✓ | ✓ | 26 |

- **Rubric-based, open-ended evaluation.** We introduce outcome-based, fine-grained rubrics that provide rigorous evaluation of long-form research answers and closely align with expert judgments. We also separate mandatory (required for sufficiency) from optional criteria, addressing a key gap in existing benchmarks.

- **Ternary Grading.** We propose a ternary grading scheme for a rubrics-based benchmark that supports partial credit assignment, and examine its suitability for automated evaluation.

- **Rubric design impact on LLM-as-a-judge.** We introduce practical recommendations for rubric design that improve agreement with human evaluators and are validated through ablation studies.

By releasing RESEARCHRUBRICS, we aim to catalyze progress toward trustworthy, well-justified DR assistants for complex, open-ended research tasks in a multitude of domains.

## 2 RELATED WORK

Early benchmarks have largely taken two approaches: deriving or constructing tasks from static corpora or relying on expert-curated questions.

**Derived Benchmarks** AcademicBrowse Zhou et al. (2025) and BrowseComp Wei et al. (2025) assess retrieval from academic papers or the web, while ResearchBench Liu et al. (2025) builds complex queries from static data. More recent work goes further and derives tasks from dynamic, real-world scenarios. DeepScholar-Bench Patel et al. (2025) evaluates systems on related work writing using live queries from arXiv papers, though it is specialized to academic synthesis and uses automated metrics. ReportBench Li et al. (2025) leverages published surveys as ground truth, measuring overlap with expert-written reviews but prioritizing replication. DeepResearch Arena Wan et al. (2025) automatically curates 10,000 open-ended tasks from academic seminars, pairing them with adaptively generated rubrics, though automatic rubric generation can miss domain nuances.

**Expert Curated Benchmarks** Expert-authored benchmarks include Humanity's Last Exam (HLE) Phan et al. (2025), which provides expert-written short-answer questions across advanced domains, but does not target more ambiguous/open-ended analysis directly, and DeepResearch Bench Du et al. (2025), which introduces 100 PhD-level problems requiring long-form reports, but struggles with critical weaknesses including LLM-generated rubrics for specialized domains, evaluation metrics reliant on LLM-generated reference reports, and simplistic reference overlap metrics. Newer works such as Mind2Web2 Gou et al. (2025) and ResearcherBench Xu et al. (2025) extend this approach, but in an effort to ease evaluation, either target narrowly defined domains (e.g., only AI-related topics) or restrict the scope of the prompt so that the Agent-as-a-Judge framework can operate effectively with LLM-generated rubrics. LiveResearchBench Wang et al. (2025) tries to focus on more realistic prompts but still relies on LLM-generated rubrics that are only human-reviewed, which may introduce an anchoring bias. Our rubrics, by comparison, are fully human-written and reviewed from the outset. ExpertLongBench Ruan et al. (2025) is similar to our benchmark in that it targets expert-level, long-form tasks across nine domains with domain-specific rubrics. However, it relies on high-quality existing references for evaluation using the CLEAR framework, which limits

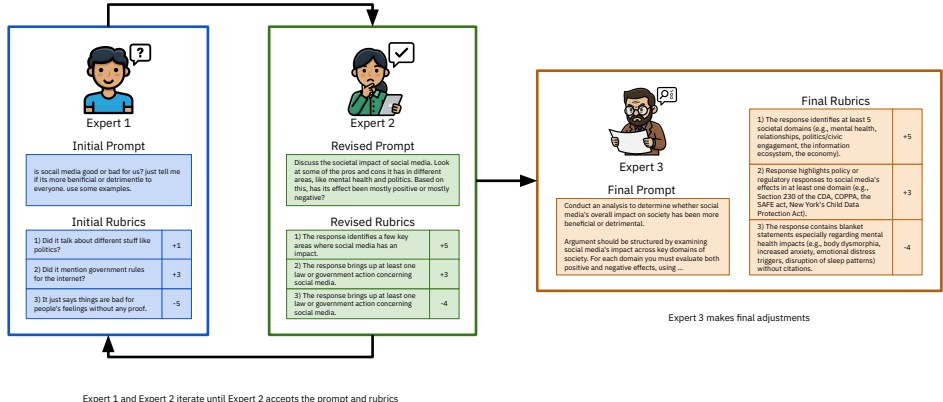

Figure 2: The three-stage pipeline for creating and refining prompts and rubrics. An initial draft by Expert 1 is iteratively improved with Expert 2 before a final review and adjustment by Expert 3.

the scope of prompts to highly academic or professional queries (e.g., Clinical Note Generation, Legal Multi-Document Case Summarization, Molecule Description Generation), whereas we also include general consumer research queries. Additionally, our benchmark has a much higher average number of rubrics per task than many of these benchmarks, resulting in superior evaluation granularity.

To summarize, existing benchmarks thus face two primary limitations: a reliance on static datasets or answer keys Liu et al. (2025); Li et al. (2025), and the use of non-expert or automated evaluation, including coarse metrics Patel et al. (2025) or auto-generated rubrics Wan et al. (2025); Du et al. (2025); Gou et al. (2025). In contrast to these approaches, RESEARCHRUBRICS offers a middle ground: realistic research queries (academic and everyday domains) paired with expert-written rubrics assessing grounding, synthesis, reasoning, clarity, and citation usage. By using human-written rubrics with LLM judges, we avoid simplistic overlap measures while maintaining scalability. RE-SEARCHRUBRICS complements efforts like ExpertLongBench, while emphasizing domain diversity, rubric quality, and a focus on deep-research specific tasks.

## 3   OVERVIEW OF RESEARCHRUBRICS

RESEARCHRUBRICS consists of 101 single-turn prompts, each paired with a set of 20–43 prompt-specific rubric criteria. Every prompt and criterion in RESEARCHRUBRICS was written and iteratively refined by human experts to ensure clarity and relevance (no criteria were seeded or generated by LLMs). The prompts cover a wide range of topics and inquiry types to emulate real user questions that deep research agents receive. In total, the benchmark contains 2,593 unique rubric items, enabling a fine-grained assessment of open-ended, realistic research queries. Figs. 1 and 2 provide an overview of our benchmark design and evaluation process.

### 3.1   DATA COLLECTION AND TASK DOMAINS

Our data collection pipeline consists of three expert participants, as shown in Fig. 2. In this context, we define an "expert" as an individual with a strong STEM background who is skilled in task design and evaluation, rather than a domain-specific specialist for each prompt. All participants in our data collection only chose and worked on domains they were familiar with.

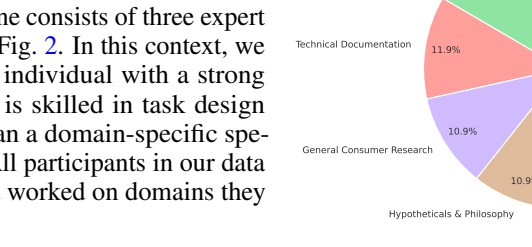

Figure 3: Distribution of task domains in our collected data.

The pipeline involves three experts, each assigned to a distinct and separate role. Expert 1 initially proposes a prompt and a set of rubric criteria. This proposal is then passed to Expert 2 for review. Expert 2 provides feedback and iterates with Expert

1 until the pair is approved. Finally, Expert 3 conducts a final, independent review and makes any last adjustments. This three-participant setup ensures that each component is thoroughly reviewed multiple times, guaranteeing high quality in the final data.

To ensure realism and variety, initial prompt ideas were drawn from user forums, Q&A sites, and brainstorming sessions, then adapted to represent the range of research-like questions a deep reasoning agent might encounter. The result is a collection of prompts that span both **breadth** (a wide variety of domains) and **depth** (challenging multi-step problems).

For each finalized prompt, experts developed a detailed rubric specifying what an ideal response should include and which common errors to avoid, following the pipeline detailed in Fig. 2. We weighted each criterion based on its importance (see Section 3.3) and included negative criteria targeting likely pitfalls, such as factually incorrect statements, off-topic tangents, or disallowed content.

We curated prompts from **nine broad categories** (see Table 10 in the Appendix for a detailed description of each category) to maximize diversity. These range from technical documentation to historical analysis, creative writing, and current events.

Fig. 3 shows the distribution of categories in RESEARCHRUBRICS. The distribution is fairly even, with AI/ML and historical analysis queries constituting the largest portions closely, followed by domains like general consumer research, reflecting both specialized academic topics and everyday research questions. Other categories provide targeted challenges (e.g., creative synthesis or real-time news retrieval). This diversity ensures that a DR agent must draw on a wide range of knowledge sources and adapt to different task structures.

## 3.2 PROMPT COMPLEXITY DIMENSIONS

Not all research prompts are equal—some involve a broader knowledge base, others require deeper reasoning, and others are underspecified and exploratory. We categorize each RESEARCHRUBRICS task along three orthogonal complexity dimensions: **Conceptual Breadth**, **Logical Nesting Depth**, and **Exploration** (Table 2). This framework helps ensure our benchmark covers a balanced mix of task types and allows analysis of where agents struggle most.

Table 2: Prompt complexity categories used to annotate each task in RESEARCHRUBRICS.

| Complexity Axis | Level | Definition | Example |
|---|---|---|---|
| **Conceptual Breadth** | *Simple* | Involves a single domain or topic; solvable using 1 primary information source or conceptual framework. | A math word problem or a factual lookup from one source. |
| | *Moderate* | Integrates 2–5 distinct subtopics or data sources that are weakly coupled; limited cross-domain reasoning. | A prompt combining two fields (e.g., a physics concept applied in a medical device context). |
| | *High* | Requires synthesis across > 5 information sources or clearly disjoint domains (e.g., science, economics); reasoning depends on multiple perspectives. | "Analyze the environmental, economic, and political factors affecting renewable energy adoption in Asia." |
| **Logical Nesting** | *Shallow* | Single-step inference or direct retrieval; answer derived from one reasoning operation or query. | "What is the capital of X country?" or a single lookup query. |
| | *Intermediate* | Multi-step reasoning (2 to 3 dependent sub-questions) where later steps depend on earlier intermediate results. | "Find the sales of Company A and Company B last year and determine who grew faster; then identify one reason for that difference." |
| | *Deep* | Requires 4+ dependent reasoning steps or hierarchical planning (e.g., analysis → synthesis → evaluation → revision). | "Develop an evidence-backed investment strategy given current economic indicators, stress-test it against at least two historical scenarios and suggest contingency plans." |
| **Exploration** | *Low* | Fully specified and unambiguous; prompt contains explicit goals, constraints, and evaluation criteria. | "Summarize the methodology of the referenced paper." The task is clear-cut. |
| | *Medium* | Moderately open-ended (1–2 unspecified factors); requires limited prioritization among known aspects. | "Discuss the benefits and risks of AI in healthcare." Covers standard themes (privacy, accuracy, etc.). |
| | *High* | Underspecified or exploratory; 3+ key factors unspecified, requiring clarification of objectives or creative reframing. | "I want to change careers to something with strong future growth—what should I consider?" The agent must clarify the criteria and explore multiple paths. |

Every task in RESEARCHRUBRICS is annotated with a triplet of (Breadth, Depth, Ambiguity) labels. In our evaluations, we analyze model performance across these dimensions to see, for example, if a model struggles more with breadth (integrating many sources) or with depth (long reasoning chains). This also helps researchers filter the benchmark for specific experiment focuses (e.g., testing only high-depth reasoning tasks).

## 3.3 RUBRIC DESIGN

RESEARCHRUBRICS is a rubric-based benchmark: each prompt is judged against a tailored set of criteria that define the requirements of a good answer. RESEARCHRUBRICS also separates mandatory (required for sufficiency) from optional criteria, addressing a key gap in existing benchmarks.

Table 3: Rubric criteria used to evaluate responses, with illustrative examples for each category.

| Criterion | Description | Example |
|---|---|---|
| **Explicit Requirements** | Checks whether the answer addresses all points explicitly asked in the prompt and does so correctly. | Prompt: "Compare X and Y and recommend one." → The answer compares X vs. Y on relevant traits and makes a clear recommendation. |
| **Implicit Requirements** | Covers points that a well-informed person would expect, even if not directly asked. Encourages completeness and contextual understanding. | Prompt: "Explain a medical treatment." → A good answer also mentions side effects or costs, even if not requested. |
| **Synthesis of Information** | Evaluates whether the model connects and synthesizes information across multiple sources or sub-parts of the query, rather than merely listing facts. | Prompt: "Summarize several studies on renewable energy adoption." → The answer identifies overarching trends and draws integrated conclusions. |
| **Use of References** | Assesses inclusion and appropriateness of citations or evidence where expected. Checks if references are specific, relevant, and actually support claims. | Prompt: "Summarize recent findings on large language models." → The answer cites key papers (e.g., "Attention is All You Need") and links claims to sources. |
| **Communication Quality** | Evaluates clarity, organization, and tone. A response may be factually correct but still poor if disorganized or misaligned with the audience's needs. | Prompt: "Write a short blog post for a general audience." → The answer is logically structured, concise, and avoids excessive jargon. |
| **Instruction Following** | Checks adherence to explicit user instructions or constraints (e.g., required format, tone, exclusions). | Prompt: "Summarize this without mentioning Topic Z." → The answer omits Topic Z as instructed. |

Table 3 presents the six broad **evaluation axes** used to assess response quality. Each axis contains multiple rubric criteria, which are categorized as either **mandatory** or **optional**.

- **Mandatory** criteria define the minimum requirements for a valid response, i.e., core elements that must be satisfied for the answer to be considered correct or adequate.

- **Optional** criteria capture desirable but non-essential qualities ("nice-to-have" behaviors) that distinguish strong responses from merely sufficient ones.

Each criterion is assigned a numerical weight in the range $[-5, 5]$, reflecting its relative importance. Weights of $\pm 4$ or $\pm 5$ correspond to mandatory criteria, while criteria with weights in $[-3, 3]$ are optional. Positive weights reward the presence of valuable attributes, while negative weights penalize common failure modes such as factual inaccuracies, irrelevance, or verbosity. These weights are aligned with a calibrated **human preference scale** (Table 7) spanning six levels, from *Critically Detrimental* to *Critically Important*. This mapping encourages more consistent human–model agreement during grading.

## 3.4 EVALUATION METHODOLOGY

Each model response is evaluated against all the rubric criteria using a model as a grader, in an LLM-as-a-judge setup. The model-based grader outputs ternary judgment verdicts for each rubric, which are {Satisfied, Partially Satisfied, Not Satisfied}. This scoring process is the same for negative criteria, which are phrased so that the negative weights are applied to the sum if the negative criteria are met. The final task score is the weighted sum of all positive and negative weights, normalized by sum of the positive weights (the maximum possible score the model can achieve).

$$S_k = \frac{\sum_{r_i \in C} w_{r_i} m_{r_i}}{\sum_{r_i \in C,\, w_{r_i} > 0} w_{r_i}}, \quad m_{r_i} = \mathrm{Judge}(P_k, \mathrm{Res}, r_i) = \begin{cases} 1, & \text{if } r_i \text{ is satisfied,} \\ 0.5, & \text{if } r_i \text{ is partially satisfied,} \\ 0, & \text{if } r_i \text{ is not satisfied,} \end{cases} \quad (1)$$

where $S_k$ is the final task score for the task $k$ with prompt $P_k$ and model response $\mathrm{Res}$. $C$ is the set of all criteria, $w_{r_i}$ is the (possibly negative) weight assigned to criterion $r_i$,, and $m_{r_i}$ is the ternary indicator returned from the model-based judge, $\mathrm{Judge}(\cdot, \cdot, \cdot)$, representing the level of satisfaction for criterion $r_i$.

To calculate the breakdown of failures per rubric category in an average task, we employ the following formula (where a failure is only when a rubric receives a `Not Satisfied` verdict).

$$\overline{F}_c = \frac{1}{|T_c|} \sum_{t \in T_c} f_{c,t} = \frac{1}{|T_c|} \sum_{t \in T_c} \frac{n_{\text{fail},c,t}}{n_{\text{fail},t}} \tag{2}$$

where $n_{\text{fail},c,t}$ is the number of failed rubrics from category $c$ in task $t$, $n_{\text{fail},t}$ is the total number of failed rubrics across all categories in task $t$, $f_{c,t}$ is the failure rate of category $c$ within task $t$, $T_c$ is the set of tasks in which category $c$ occurs at least once, and $\overline{F}_c$ is the average failure rate of category $c$ across tasks.

This allows us to understand that when rubrics fail, which categories are responsible for the highest contribution of failures in an average task (as opposed to just how often rubrics from a certain category fail). An important feature to note is that since the failure rate breakdown is averaged across only those tasks in which those rubric categories occur (to minimize the effect of an imbalanced rubric category distribution), the failure rate ratios do not necessarily add up to 1.

**Human Consistency Analysis**  Similar to HealthBench Arora et al. (2025), we utilize the Macro $F1$ score to validate the effectiveness of using a model-based grader as a proxy for human judgment. In our setup, we compare the ground truth judgement of experts and model-based graders for each task, and compute the $F1$ scores for each of the classes {`Satisfied`, `Partially Satisfied`, `Not Satisfied`}.

$$F_1 = 2 \cdot \frac{\text{precision} \cdot \text{recall}}{\text{precision} + \text{recall}}, \text{ where precision} = \frac{TP}{TP+FP} \text{ and recall} = \frac{TP}{TP+FN}, \tag{3}$$

where $TP$, $FP$, and $FN$ are the True Positive, False Positive, and False Negative values, respectively. We also run ablation studies to isolate the most significant factors in the level of alignment between the model-based grader and human judgments. For more details, see Section 4.4.

## 4 EXPERIMENTAL RESULTS AND ANALYSIS

We evaluate three commercial Deep Research (DR) agents on RESEARCHRUBRICS to measure their capabilities across multi-step synthesis, implicit reasoning, and evidence-backed justification. Our benchmark introduces 2,500+ expert-written rubric criteria across 100+ prompts, providing a more granular evaluation than existing frameworks. This granularity enables atomic-level quality assessment that allows us to identify specific failure modes invisible to coarse-grained metrics.

### 4.1 EXPERIMENTAL SETUP

**Evaluated Systems**  We benchmark OpenAI Deep Research OpenAI (2025a), Gemini Deep Research Google (2025), and Perplexity Deep Research AI (2025). Each system produces structured PDF reports that we convert to markdown for evaluation across six dimensions: Explicit Requirements, Implicit Reasoning, Synthesis of Information, References, Communication Quality, and Instruction Following. Our evaluation employs both binary (`met`/`not-met`) and ternary (`fully`/`partially`/`not satisfied`) grading schemes to understand the impact of partial credit on system rankings.

**LLM-as-Judge Implementation**  We deploy three state-of-the-art LLMs as automated judges: GPT-5 OpenAI (2025b), Claude-Sonnet-4.5 Anthropic (2025), and Gemini-2.5-Pro DeepMind (2025). Under binary grading, we collapse `Partially Satisfied` verdicts to `Not Satisfied`, measuring strict compliance. Human–model alignment is quantified using Macro $F_1$ scores, with nine expert annotators providing ground truth across 303 responses.

### 4.2 MAIN RESULTS

**Compliance Scores**  Table 4 reveals that **no current system exceeds 70% rubric compliance**, with the best-performing Gemini DR achieving only 67.7% under ternary grading and

Table 4: **Overall Compliance Scores**

| Model | Ternary | Binary |
|---|---|---|
| Gemini DR | **0.677** | **0.615** |
| OpenAI DR | 0.664 | 0.597 |
| Perplexity DR | 0.566 | 0.487 |

61.5% under binary evaluation. This aligns with findings from LiveResearchBench, where leading systems score below 74% on comprehensive metrics, DeepResearch Bench, where leading systems score below 50% on comprehensive metrics. The consistency across benchmarks suggests fundamental architectural limitations rather than benchmark-specific challenges.

**Failure Rates**    Fig. 4 decomposes failure rates across evaluation dimensions, revealing that **implicit reasoning and synthesis jointly account for 45-50% of all failures**. This corroborates the findings in Multi-Agent System Taxonomy (MAST) Cemri et al. (2025), identifying reasoning-action mismatch (13.98%) and disobedience of task specifications (10.98%) as systemic issues. While agents excel at explicit factual retrieval and communication quality (failure rates below 20%), they consistently fail to infer unstated requirements or integrate multi-document evidence into coherent arguments.

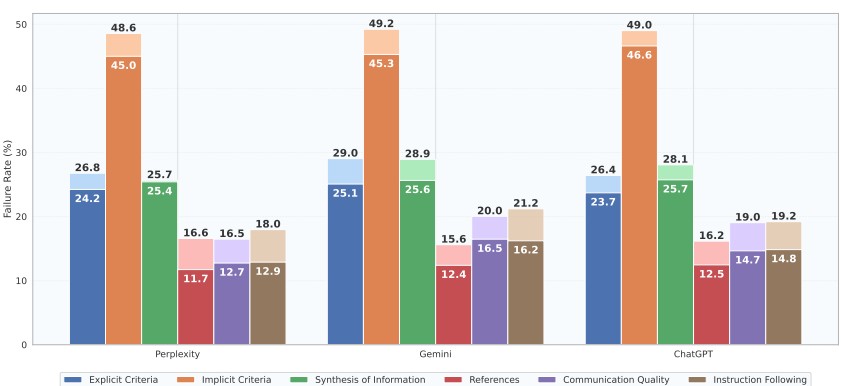

Figure 4: **Rubric-axis failure rates across Deep Research agents.** Dark bars represent ternary grading; light bars show binary grading. Implicit reasoning and synthesis show markedly higher failure rates compared to communication quality and references. The pattern holds across all three systems, indicating architectural rather than implementation limitations.

**Mandatory vs. Optional Criteria**    RESEARCHRUBRICS separates mandatory and optional criteria, and using this differentiation, we observe (from Fig. 6) that, while mandatory criteria drive failures in explicit requirements and synthesis of information, optional criteria account for most implicit reasoning failures. This suggests current systems meet basic implicit requirements but miss nuanced quality indicators that distinguish professional from adequate research.

This finding contextualizes HealthBench's worst-at-16 analysis showing 33% performance degradation from average to minimum—systems achieve moderate average scores by satisfying mandatory criteria while systematically missing optional quality dimensions. The mandatory/optional distinction proves essential for deployment decisions: a 60% overall score might indicate either dangerous gaps in core requirements or merely missing polish on otherwise solid foundations.

**Performance Stratified by Complexity Dimension**    Fig. 5 presents model compliance scores stratified by conceptual breadth, logical nesting, and exploration level under binary and ternary grading schemes, respectively. Gemini DR consistently leads, achieving roughly 70% average rubric compliance across most complexity tiers, followed closely by ChatGPT DR, and Perplexity DR lagging slightly behind. A clear pattern emerges: performance degrades monotonically with increased logical nesting depth. Whereas shallow reasoning tasks (single-hop or two-step queries) are handled well, multi-step analytical or evaluative problems see sharp drops, particularly for models relying on retrieval-centric architectures. Conceptual breadth also correlates with difficulty, though less steeply; systems handle multi-domain synthesis better than extended inferential chaining.

**Citation Analysis**    The implicit reasoning gap explains the breadth-accuracy trade-off documented in citation analysis: Gemini DR produces 111 citations with 81% accuracy while Perplexity achieves 90% accuracy with only 31 citations. Systems optimized for comprehensive coverage sacrifice precision, while those targeting accuracy miss crucial perspectives—neither strategy successfully handles the implicit judgment of source relevance and authority.

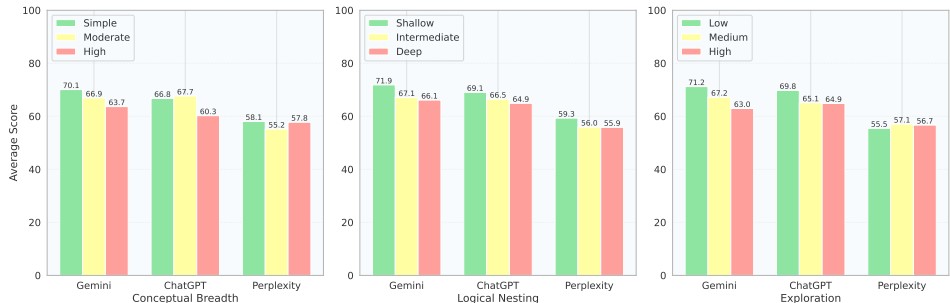

Figure 5: Performance across Conceptual Breadth, Logical Nesting, and Exploration (Ternary Evaluation)

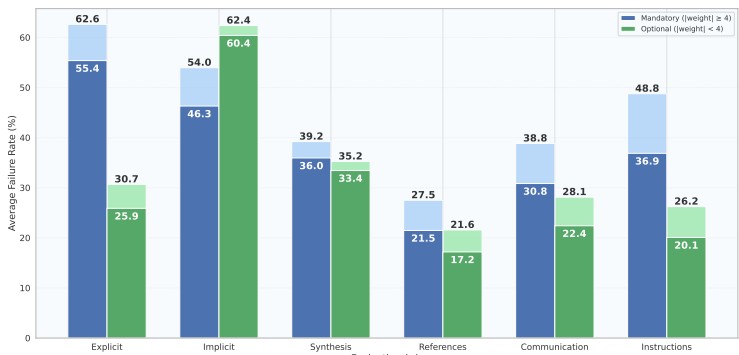

Figure 6: **Failure rate stratification by criterion importance.** Mandatory criteria show systematically higher failure rates across most dimensions, with the notable exception of implicit reasoning, where optional criteria failures dominate. This inversion suggests implicit requirements primarily distinguish excellent from merely sufficient responses. Dark bars represent ternary grading; light bars show binary grading.

## 4.3    HUMAN-LLM JUDGE ALIGNMENT FOR AUTO-EVALUATION

Our human evaluation study (Table 5) demonstrates that binary grading achieves substantial agreement (0.72–0.76 Macro $F_1$), approaching the best-performing LLM-judges for rubrics benchmarks in recent literature. The shift from ternary to binary evaluation increases agreement by approximately 20 percentage points, confirming that partial credit introduces ambiguity without improving discriminative power.

Table 5: **Human consistency with LLM judges.** Macro $F_1$ scores between human annotators and automated evaluation across grading schemes and judge models.

| | Agent | Judge Model Agreement | | |
|---|---|---|---|---|
| | | GPT-5 | Claude-4.5 | Gemini-2.5-Pro |
| *Binary* | Perplexity DR | 0.717 | 0.718 | **0.724** |
| | Gemini DR | 0.732 | 0.741 | **0.760** |
| | OpenAI DR | 0.719 | **0.742** | 0.721 |
| *Ternary* | Perplexity DR | 0.538 | 0.528 | **0.559** |
| | Gemini DR | 0.553 | 0.532 | **0.567** |
| | OpenAI DR | 0.546 | 0.527 | **0.557** |

The consistency levels validate automated evaluation feasibility for RESEARCHRUBRICS's 2,593 criteria, exceeding HealthBench's 0.709 Macro $F_1$ score. Gemini-2.5-Pro emerges as the most reliable judge, achieving 0.76 agreement on binary grading, though at least the 12-17 percentage point gap to best human agreement indicates remaining room for improvement.

## 4.4 RUBRIC DESIGN IMPACT

To better understand how rubric design impacts evaluation reliability, we conducted a series of **ablation studies** focusing on two key factors: (1) the inclusion of concrete examples within rubric criteria, and (2) the use of LLM-based augmentation to automatically rephrase those criteria. The goal of these experiments was to measure how such modifications affect alignment between automated (LLM-as-judge) and human evaluations. We present the results of the ablation study in Table 6.

We began with the original, expert-authored rubrics as our control condition. *Example Detail* tests whether providing brief, inline examples for each criterion improves agreement between human and model judges (in the format "(e.g., example1, example2, example3)"). The "Low" condition uses minimal guidance (the baseline criteria only), whereas "High" includes short, task-relevant examples (e.g., a cited study, policy name, relevant item). *LLM Augmentation* evaluates whether prompting a large language model to automatically expand or rephrase rubric text adds clarity. In the "Absent" setting, rubrics are the original human-written ones; in the "Present" setting, each rubric was rewritten by an LLM with added qualifiers and examples.

We find, in Table 6, that including concrete examples within rubric criteria improves alignment by 3-4% (binary) and 2-3% (ternary). However, LLM-based rubric augmentation, i.e., automatically expanding criteria with synthetic elaboration, **catastrophically degrades alignment by 15-20%**.

Table 6: **Impact of rubric design on evaluation reliability.** Adding examples improves human-LLM alignment while automated augmentation degrades it.

|  | Agent | Example Detail | | LLM Augmentation | |
|---|---|---|---|---|---|
|  |  | Low | High | Absent | Present |
| *Binary* | Perplexity DR | 0.696 | 0.724 | 0.724 | 0.508 |
|  | Gemini DR | 0.733 | 0.760 | 0.760 | 0.564 |
|  | OpenAI DR | 0.709 | 0.721 | 0.721 | 0.528 |
| *Ternary* | Perplexity DR | 0.523 | 0.559 | 0.559 | 0.371 |
|  | Gemini DR | 0.539 | 0.567 | 0.567 | 0.417 |
|  | OpenAI DR | 0.532 | 0.557 | 0.557 | 0.387 |

This finding challenges assumptions about verbosity improving clarity. Human-authored concise rubrics with targeted examples outperform machine-generated verbose descriptions, likely because augmentation introduces semantic drift and emphasis distortion. The implication for RE-SEARCHRUBRICS' 2,593 criteria is clear: **expert curation cannot be replaced by automated expansion, and clarity emerges from precision rather than elaboration**.

## 5 CONCLUSION AND FUTURE WORK

We introduced RESEARCHRUBRICS, a new benchmark and evaluation framework for deep research agents that emphasizes fine-grained, human-aligned assessment. Through 101 diverse research challenges and expert-written rubric criteria, our benchmark provides a multi-dimensional lens on an agent's performance—checking not just factual recall, but the completeness, reasoning soundness, source usage, and clarity of its responses. RESEARCHRUBRICS 's granularity enables us to identify specific capability gaps invisible to aggregate metrics, and the mandatory/optional distinction gives us a way to place an agent on the sufficiency–excellence continuum, aiding deployment decisions by focusing on minimum viable performance rather than average scores. Our experiments reveal that today's best agents achieve only around 67% compliance with these rigorous rubrics, often falling short in integrating information across documents and providing well-justified answers with proper citations. Most critically, our findings suggest that improving Deep Research agents requires architectural innovation rather than incremental refinement: systematic failures in implicit reasoning, multi-document synthesis, and sustained sequential reasoning point to fundamental limitations in how current systems represent and manipulate complex information structures.

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

## A    EXTENDED RELATED WORK

The rapid emergence of deep research agents has been accompanied by several efforts to characterize and evaluate their capabilities. Recent surveys and roadmap papers highlight the promise and challenges of autonomous LLM-based research assistants. For example, Huang et al. (2025) provide a systematic examination of Deep Research agents, analyzing their tool integration and planning strategies, while Xu and Peng (2025) offer a comprehensive survey of deep research systems and applications. These works underscore the need for robust evaluation frameworks aligned with the complex, open-ended nature of research tasks.

Early benchmarks for deep research agents have largely taken one of two approaches: constructing tasks from static corpora or relying on expert-curated questions. In the first category, benchmarks like **AcademicBrowse** Zhou et al. (2025) and **BrowseComp** Wei et al. (2025) assess an agent's ability to navigate and retrieve information from academic papers or the web. AcademicBrowse focuses on literature-based queries (e.g., browsing academic papers for answers), and BrowseComp comprises over 1,200 web questions that demand multi-hop searching across sites. While these benchmarks test long-horizon retrieval and factual accuracy, their questions tend to have a predetermined scope or "ground truth" answers, which simplifies evaluation to matching reference facts. This limits their ability to capture the open-ended synthesis and exploratory aspect of real research inquiries. Another example is **ResearchBench** Liu et al. (2025), which builds complex search questions from static data; however, static benchmarks risk *data leakage* (i.e., answers appearing in training data) and cannot adapt to newly emerging information.

The second category of benchmarks uses expert-authored tasks to evaluate research reasoning. **Humanity's Last Exam** (HLE) Phan et al. (2025) is an expansive evaluation of 2,500 expert-written questions covering advanced domains ranging from mathematics to medicine. HLE revealed significant gaps in state-of-the-art models' knowledge, but it primarily consists of challenging short-answer questions, rather than multi-document analytical tasks. Closer to our setting, **DeepResearch Bench** Du et al. (2025) introduced 100 PhD-level research problems across 22 fields (e.g., scientific analysis, legal reasoning), each requiring a long-form report. Their evaluation combines reference-based metrics and adaptive criteria, including measuring the number and accuracy of citations. This benchmark confirmed the difficulty of deep research tasks, where no model exceeded roughly 30% on their overall metrics, yet its scoring approach leans heavily on overlap with reference solutions and simple citation counts. Similarly, **ExpertLongBench** Ruan et al. (2025) targets expert-level, long-form tasks in 9 domains (law, finance, healthcare, etc.), providing 11 complex prompts each accompanied by a domain-specific checklist or rubric. ExpertLongBench introduced the CLEAR evaluation framework, which extracts a structured checklist from both the model's output and a gold reference, then compares them for alignment. This method enables fine-grained assessment of content requirements, but it depends on high-quality reference outputs for each task. In contrast, our work uses expert-written criteria without assuming an ideal reference answer, and evaluates responses directly via LLM-as-a-judge – avoiding potential biases from any single ground-truth essay.

More recent benchmarks have moved toward dynamic, real-world research scenarios. **DeepScholar-Bench** Patel et al. (2025) focuses on *generative research synthesis*: it draws live queries from recent arXiv papers and evaluates systems on writing a related work section by retrieving and summarizing up-to-date literature. Its evaluation emphasizes three axes (knowledge synthesis, retrieval quality, and verifiability), rewarding comprehensive coverage of relevant work and correct citation of sources. However, DeepScholar-Bench is specialized to academic writing tasks, and uses automated metrics (including LLM-generated scores) which may introduce evaluation circularity. **ReportBench** Li et al. (2025) takes another automated approach by leveraging existing survey articles as ground truth for evaluation. It generates academic survey-style prompts and measures the overlap between the AI agent's citations and statements and those in a published survey on the same topic. This provides a concrete correctness signal (since an expert-written literature review is treated as the gold standard), but inherently prioritizes replication of the reference content over creative or divergent but valid answers. Meanwhile, **DeepResearch Arena** Wan et al. (2025) addresses the authenticity of research prompts: it automatically curates over 10,000 open-ended tasks from transcripts of academic seminars across 12 disciplines. By capturing questions that arise organically in expert discussions, DeepResearch Arena aims to evaluate agents on more ill-defined, exploratory problems. Their evaluation combines factual grounding checks with adaptively generated rubrics (checklists) to

handle the breadth of tasks. One limitation, however, is that fully automatic rubric generation can miss domain nuances or implicitly favor certain solution paths.

In parallel to benchmarking efforts, researchers have begun exploring AI "co-scientist" systems that autonomously propose hypotheses or experimental plans beyond just information retrieval. Notably, Gottweis et al. (2025) present an **AI Co-Scientist** built on a multi-agent Gemini 2.0 system, which iteratively generates and refines scientific hypotheses (demonstrated in drug discovery and biology domains). The advent of such systems raises the stakes for evaluation: beyond finding correct facts, we must assess whether an AI's reasoning and conclusions hold up to expert scrutiny. Initial work in this vein includes benchmarks like SPOT Son et al. (2025), which checks AI-generated scientific papers for logical errors or inconsistencies. Overall, as deep research agents expand from answering questions to performing nuanced scientific investigations, the need for **fine-grained, human-aligned evaluation** becomes ever more critical.

Our work builds directly on these prior insights. In contrast to previous benchmarks that either rely on static answer keys or on coarse-grained metrics, RESEARCHRUBRICS offers a new middle ground: a broad collection of realistic research queries (spanning academic and everyday domains) paired with expertly crafted rubrics that detail the requirements of a good answer. This approach enables evaluation of multiple dimensions – factual grounding, cross-source synthesis, reasoning validity, clarity, and citation usage – within a single unified framework. By using human-written rubrics and having LLM judges apply them, we avoid reward hacking based on simplistic overlap measures, while still achieving scalable scoring. RESEARCHRUBRICS is complementary to contemporaneous efforts like ExpertLongBench and DeepResearch Arena: those benchmarks target either highly specialized expert tasks or massive automatically generated task suites, whereas we prioritize diversity of domains and manually quality-checked criteria. Together, these efforts push toward a more rigorous and comprehensive assessment of deep research capabilities.

## B  EXTENDED RESULTS

This appendix expands the quantitative analysis of composition, complexity, and error structure, and clarifies the relationship between output length and rubric compliance.

### B.1  BENCHMARK COMPOSITION AND RUBRIC COVERAGE

Fig. 8 shows the number of rubric axes touched per task (mean = 4.74). This multi-axis coverage reflects our goal of measuring holistic research ability rather than single-skill performance. Fig. 9 reports the criteria count per task (20–43; mean ≈ 26). Fig. 10 decomposes axis proportions by domain, illustrating that domains differ not only by content but by the expected mix of explicitness, synthesis, and citation behaviors.

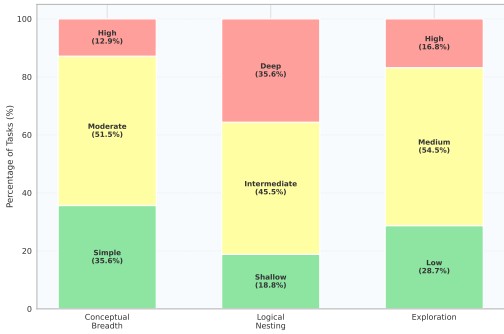

(a) Distribution of task complexity dimensions in RE-SEARCHRUBRICS.

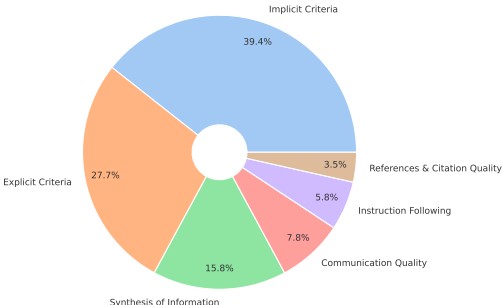

(b) Distribution of rubric criteria categories. Implicit and explicit criteria dominate the benchmark.

Figure 7: Overview of task complexity dimensions and rubric criteria category distributions in RESEARCHRUBRICS.

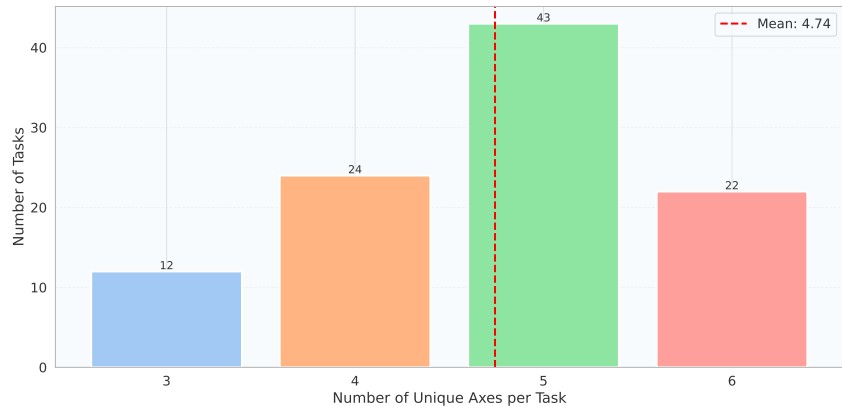

Figure 8: **How many evaluation axes does each task cover?** Distribution of the number of rubric axes per prompt. Most tasks require 4 to 5 distinct dimensions of quality simultaneously, encouraging balanced capabilities rather than single-axis optimization.

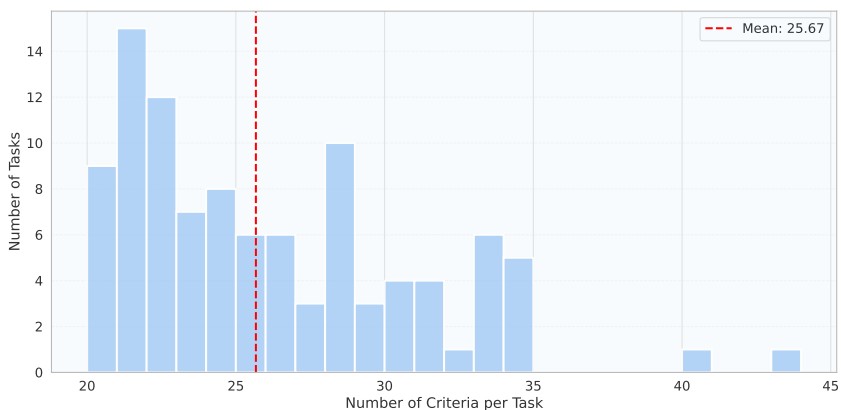

Figure 9: Number of rubric criteria per task.

## B.2 RUBRIC SCORING SCHEME

Table 7: Rubric scoring scale with mandatory and optional criteria.

| Score Range | Description |
|---|---|
| $[+4, +5]$ | **Critically important** – A criterion without which the response is fundamentally flawed or incorrect. Required for a minimally viable response. |
| $[-5, -4]$ | **Critically detrimental** – A criterion identifying an error so severe that it makes the response actively harmful, deeply unethical, or completely invalidates its reasoning. |
| $[+2 + 3]$ | **Important** – A key feature of a strong response, but not absolutely essential. |
| $+1$ | **Slightly Important** – A "nice-to-have" detail that improves a good response but does not significantly change overall quality. |
| $-1$ | **Slightly Detrimental** – A minor issue, tangent, or stylistic weakness that does not impact core reasoning or validity. |
| $[-3, -2]$ | **Detrimental** – A significant error that detracts from the response quality, introduces faulty logic, or offers poor advice, but does not make it fundamentally harmful. |

## B.3 PERFORMANCE STRATIFIED BY COMPLEXITY DIMENSION

Figs. 11 and 12 present model compliance scores stratified by conceptual breadth, logical nesting, and exploration level under binary and ternary grading schemes, respectively. Across both settings,

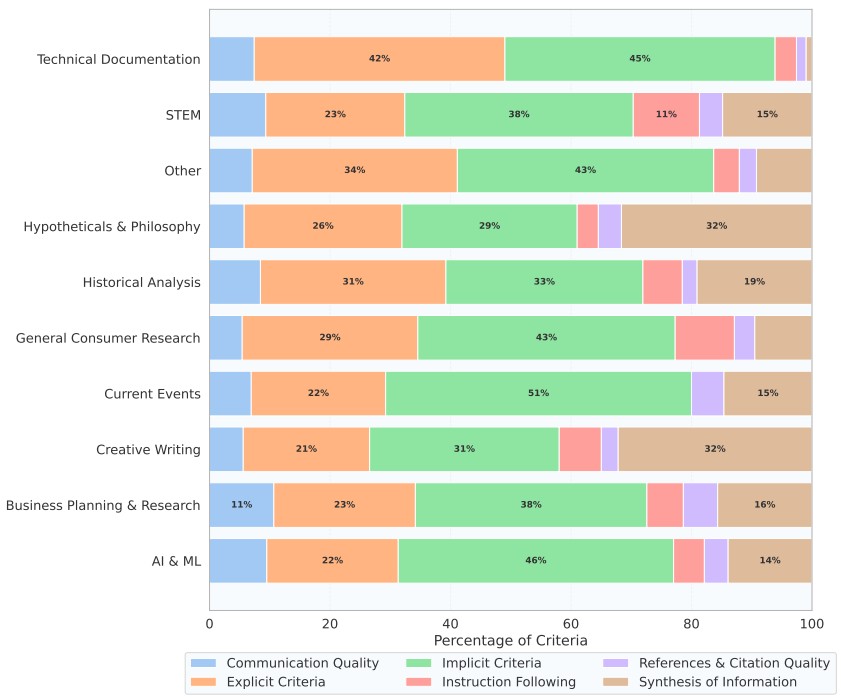

Figure 10: **Axis mix by domain.** Stacked proportions of the six rubric axes across domains.

Gemini DR consistently leads, achieving roughly 65–70% average rubric compliance across most complexity tiers, followed closely by ChatGPT DR at around 60–65%, and Perplexity DR lagging near 50%.

A clear pattern emerges: performance degrades monotonically with increased logical nesting depth. Whereas shallow reasoning tasks (single-hop or two-step queries) are handled well, multi-step analytical or evaluative problems see sharp drops, particularly for models relying on retrieval-centric architectures. Conceptual breadth also correlates with difficulty, though less steeply; systems handle multi-domain synthesis better than extended inferential chaining.

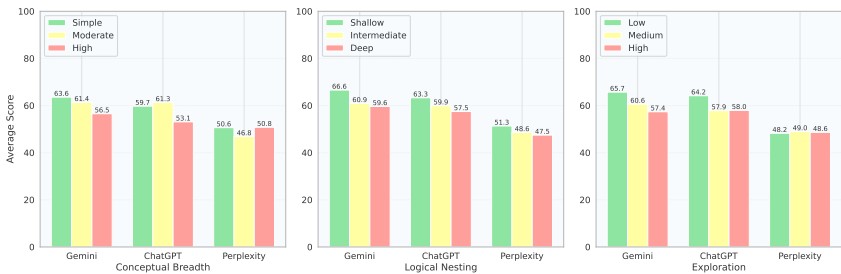

Figure 11: Performance across Conceptual Breadth, Logical Nesting, and Exploration (Binary Evaluation)

## B.4 DOMAIN-WISE FAILURE STRUCTURE

The heatmap in Fig. 13 shows how failure rates distribute across axes within each domain.

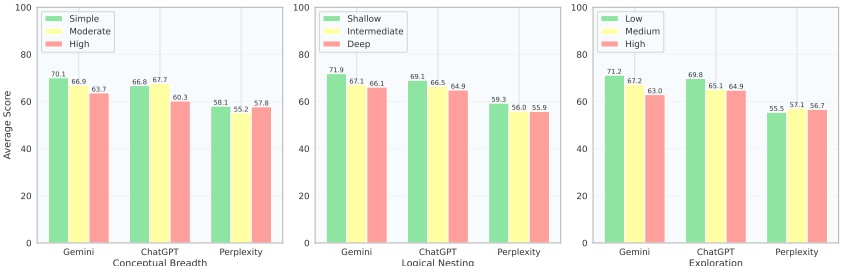

Figure 12: Performance across Conceptual Breadth, Logical Nesting, and Exploration (Ternary Evaluation)

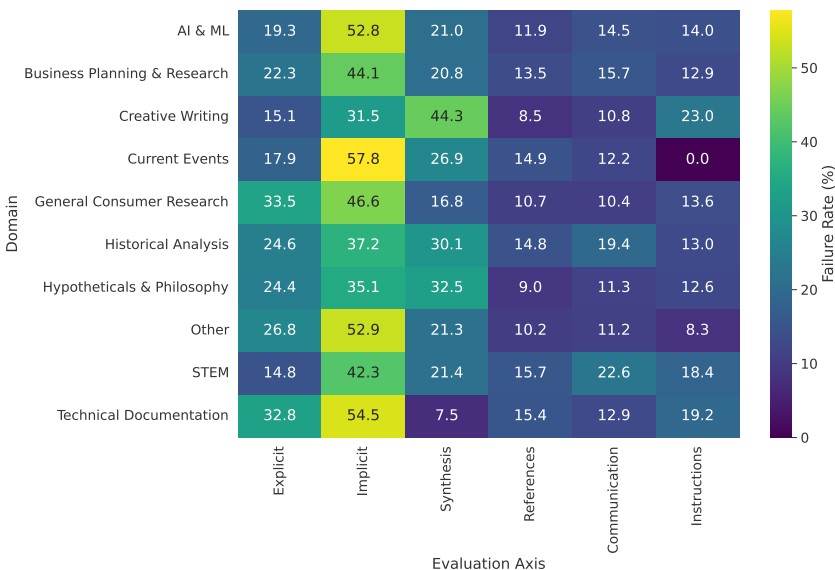

Figure 13: Heatmap of failure contribution by rubric axis across domains.

## B.5 MISCLASSIFICATION FAILURES IN HUMAN-LLM JUDGE ALIGNMENT DURING AUTO-EVALUATION

Fig. 14 illustrates the relationship between grading mismatches, i.e., disagreements between the LLM-as-a-judge and human evaluators, and various analytical dimensions across both binary and ternary classification settings. Specifically, the top row compares mismatch distributions across rubric categories, the middle row examines mismatches with respect to rubric importance (mandatory vs. optional), and the bottom row presents mismatch rates by rubric category, normalized by the size of that category in the dataset. We observe that Implicit Criteria account for the majority of misclassifications, which is unsurprising given that many rubrics in the dataset belong to this category. However, when normalized by category size, References & Citation Quality and Synthesis of Information show a slightly higher proportion of disagreements, suggesting that models may struggle to assess what constitutes an adequate mention of reference or argument in a response. We also note that mandatory criteria exhibit a lower proportion of mismatches, which is reassuring, as it implies the model and human raters tend to align more closely on the mandatory aspects of the response.

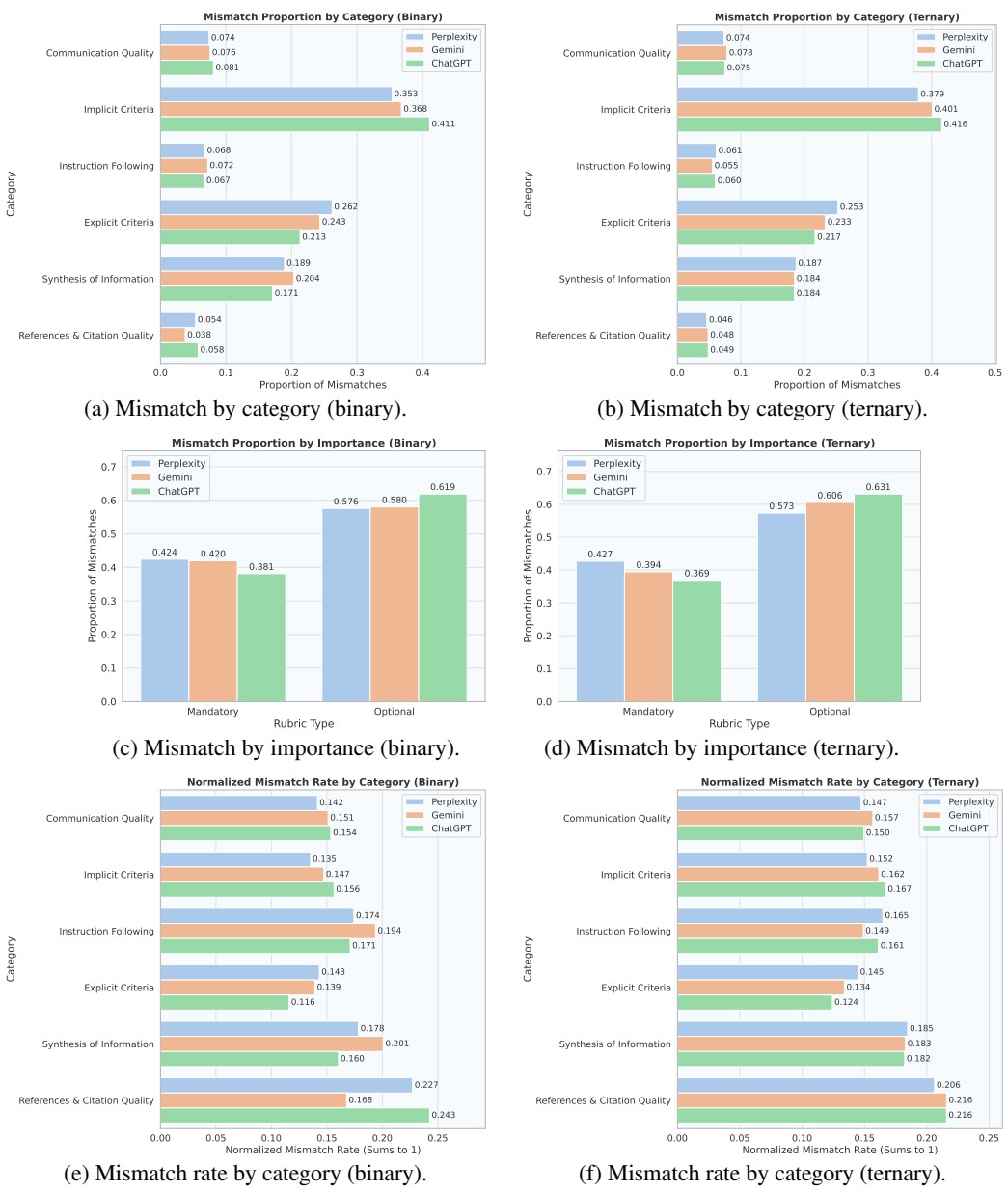

Figure 14: Comparison of mismatch metrics (by category, importance, and mismatch rate) across binary and ternary settings.

## C   PROMPT AND RESPONSE LENGTH ANALYSIS

### C.1   PROMPT WORD COUNT ANALYSIS

To understand the scope and complexity of the evaluation tasks, we analyzed the word counts of all 101 prompts included in RESEARCHRUBRICS. Prompt length serves as a useful proxy for task complexity, as longer prompts tend to encode more contextual background, sub-questions, and open-ended reasoning requirements.

Across all tasks, prompt lengths are moderately distributed, with a mean of **87.6 ± 58.6 words** (median = 68, range = 13–315). As shown in Fig. 15, most prompts cluster below 100 words, though a long right-tail distribution reflects the presence of prompts well over 200 words.

Prompts vary substantially by domain (Table 8). Tasks from **General Consumer Research**, **Technical Documentation**, and **Business Planning & Research** exhibit the longest average prompt lengths, often exceeding 100 words. In contrast, domains such as **AI & ML**, **Current Events**, and **Other** tend to be more concise.

Prompt length also scales with the benchmark's complexity dimensions (Fig. 16). Prompts with higher *conceptual breadth*, deeper *logical nesting*, and greater *exploration* are systematically longer, often doubling in average length compared to simpler tasks. This pattern underscores that more open-ended research problems require not only deeper reasoning but also more extensive prompt scaffolding.

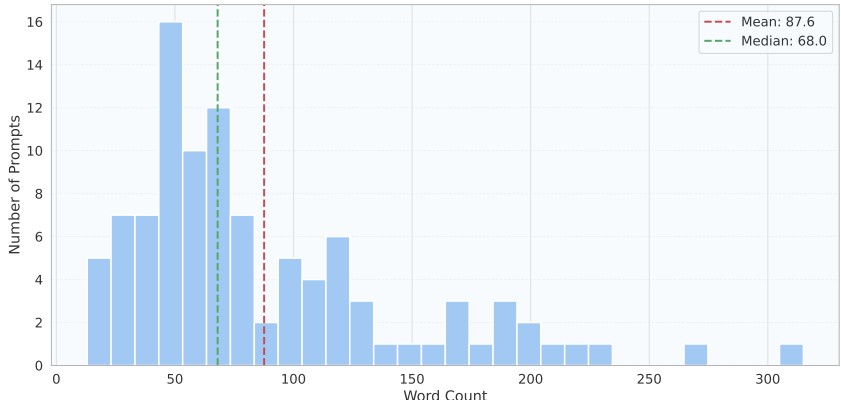

Figure 15: Distribution of prompt word counts across all 101 tasks. The distribution is right-skewed, with a mean of 87.6 words and a median of 68 words.

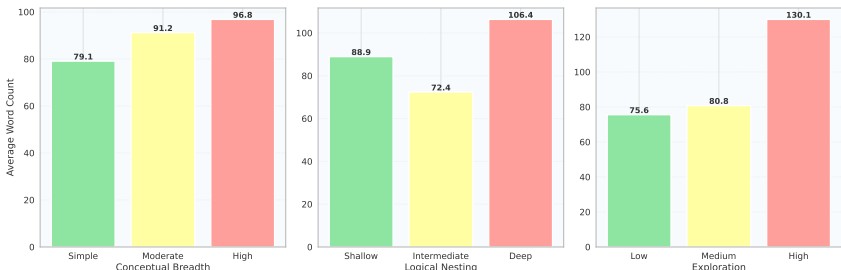

Figure 16: Prompt word count by task complexity dimensions (Conceptual Breadth, Logical Nesting, and Exploration). Longer prompts are consistently associated with higher complexity levels.

## C.2 RESPONSE LENGTH AND COMPLIANCE

To contextualize the prompt statistics, we compared the word and token counts of responses generated by three Deep Research agents: **ChatGPT DR**, **Gemini DR**, and **Perplexity DR**.

On the Markdown outputs (Table 9), **Gemini** produces the longest responses on average (7,500–7,600 words), followed by **ChatGPT** (6,300–6,400 words), while **Perplexity** outputs are substantially shorter (∼1,800 words). These differences are consistent across both words and tokens, and between text and rendered formats. High variance (standard deviations above 2,000–3,000 words) reflects substantial prompt-dependent variation in response verbosity.

To understand whether output verbosity correlates with perceived quality, we examine the relationship between response length (in tokens and words) and overall rubric compliance. Figs. 17a to 17d display these correlations for Gemini DR, ChatGPT DR, and Perplexity DR.

Moderate positive correlations ($r \approx 0.20 - 0.28$ for Gemini and ChatGPT) indicate that longer responses generally achieve higher scores. Perplexity DR, with the shortest outputs, achieves the lowest correlations.

Table 8: Prompt Word Count Statistics across Domains and Complexity Dimensions

| Category | Subset | Count | Mean | SD | Median | Min–Max | 95% CI |
|---|---|---|---|---|---|---|---|
| *Overall Statistics* | | | | | | | |
| | All Prompts | 101 | 87.6 | 58.6 | 68.0 | 13–315 | [76.0, 99.2] |
| *By Domain* | | | | | | | |
| | AI & ML | 17 | 61.8 | 44.8 | 46.0 | 13–169 | [38.0, 85.5] |
| | Business Planning & Research | 12 | 111.0 | 56.2 | 98.5 | 36–224 | [73.7, 148.3] |
| | Creative Writing | 6 | 69.2 | 19.9 | 66.0 | 40–103 | [46.2, 92.1] |
| | Current Events | 5 | 51.0 | 20.1 | 55.0 | 21–76 | [23.1, 78.9] |
| | General Consumer Research | 11 | 138.4 | 80.2 | 131.0 | 35–315 | [81.9, 194.9] |
| | Historical Analysis | 13 | 81.5 | 50.2 | 70.0 | 30–227 | [49.9, 113.1] |
| | Hypotheticals & Philosophy | 11 | 78.3 | 45.4 | 69.0 | 22–187 | [46.3, 110.2] |
| | Other | 6 | 61.2 | 22.6 | 51.0 | 40–107 | [35.2, 87.2] |
| | STEM | 8 | 80.0 | 43.9 | 64.0 | 30–174 | [40.8, 119.2] |
| | Technical Documentation | 12 | 112.3 | 69.1 | 75.5 | 49–271 | [66.5, 158.2] |
| *By Conceptual Breadth* | | | | | | | |
| | Simple | 36 | 79.1 | 58.7 | 60.5 | 13–271 | [59.0, 99.2] |
| | Moderate | 52 | 91.2 | 60.7 | 72.0 | 22–315 | [74.1, 108.3] |
| | High | 13 | 96.8 | 45.4 | 95.0 | 29–195 | [68.3, 125.4] |
| *By Logical Nesting* | | | | | | | |
| | Shallow | 19 | 88.9 | 65.8 | 67.0 | 21–227 | [56.4, 121.5] |
| | Intermediate | 46 | 72.4 | 40.4 | 61.5 | 13–197 | [60.3, 84.5] |
| | Deep | 36 | 106.4 | 68.0 | 79.5 | 22–315 | [83.0, 129.7] |
| *By Exploration* | | | | | | | |
| | Low | 29 | 75.6 | 55.1 | 56.0 | 13–227 | [54.3, 96.9] |
| | Medium | 55 | 80.8 | 48.9 | 66.0 | 21–271 | [67.5, 94.2] |
| | High | 17 | 130.1 | 72.7 | 111.0 | 47–315 | [91.5, 168.6] |

Table 9: Word and Token Statistics per Model

| Type | Model | Mean | SD | Median | Min | Max |
|---|---|---|---|---|---|---|
| Words | ChatGPT | 6269.73 | 3684.21 | 5481 | 1328 | 18824 |
| | Gemini | 7519.32 | 2447.70 | 7562 | 2909 | 14640 |
| | Perplexity | 1828.61 | 1127.70 | 1579 | 128 | 7352 |
| Tokens | ChatGPT | 10169.57 | 5885.79 | 9075 | 2103 | 30233 |
| | Gemini | 12153.31 | 4028.00 | 11710 | 4530 | 26421 |
| | Perplexity | 3664.36 | 2006.01 | 3241 | 539 | 14148 |

This supports the length–quality conflation hypothesis: longer reports often perform better because they cover more rubric criteria, not necessarily because evaluators prefer verbosity. Nonetheless, since RESEARCHRUBRICS scores are criterion-based rather than holistic, the observed correlation partly reflects genuine informational density rather than stylistic inflation.

## C.3    DISCUSSION: SYSTEMATIC PATTERNS AND THEIR IMPLICATIONS

We next present some of our key findings from our analysis.

**Domain and Task Complexity Effects**    Our analysis reveals surprising performance inversions across domains. Agents achieve 76% coverage on open-ended consulting questions but struggle with technical precision tasks, contradicting intuitive difficulty expectations. This aligns with ResearcherBench Xu et al. (2025) findings that systems excel at exploratory reasoning while failing on deterministic requirements. The pattern suggests current architectures inherently favor creative synthesis over systematic execution, explaining why even leading systems achieve below 40% on technical nugget coverage despite 85% scores on organizational structure.

Task complexity analysis confirms the depth-width decomposition framework: performance degradation accelerates with sequential reasoning requirements (depth) more than parallel capability demands

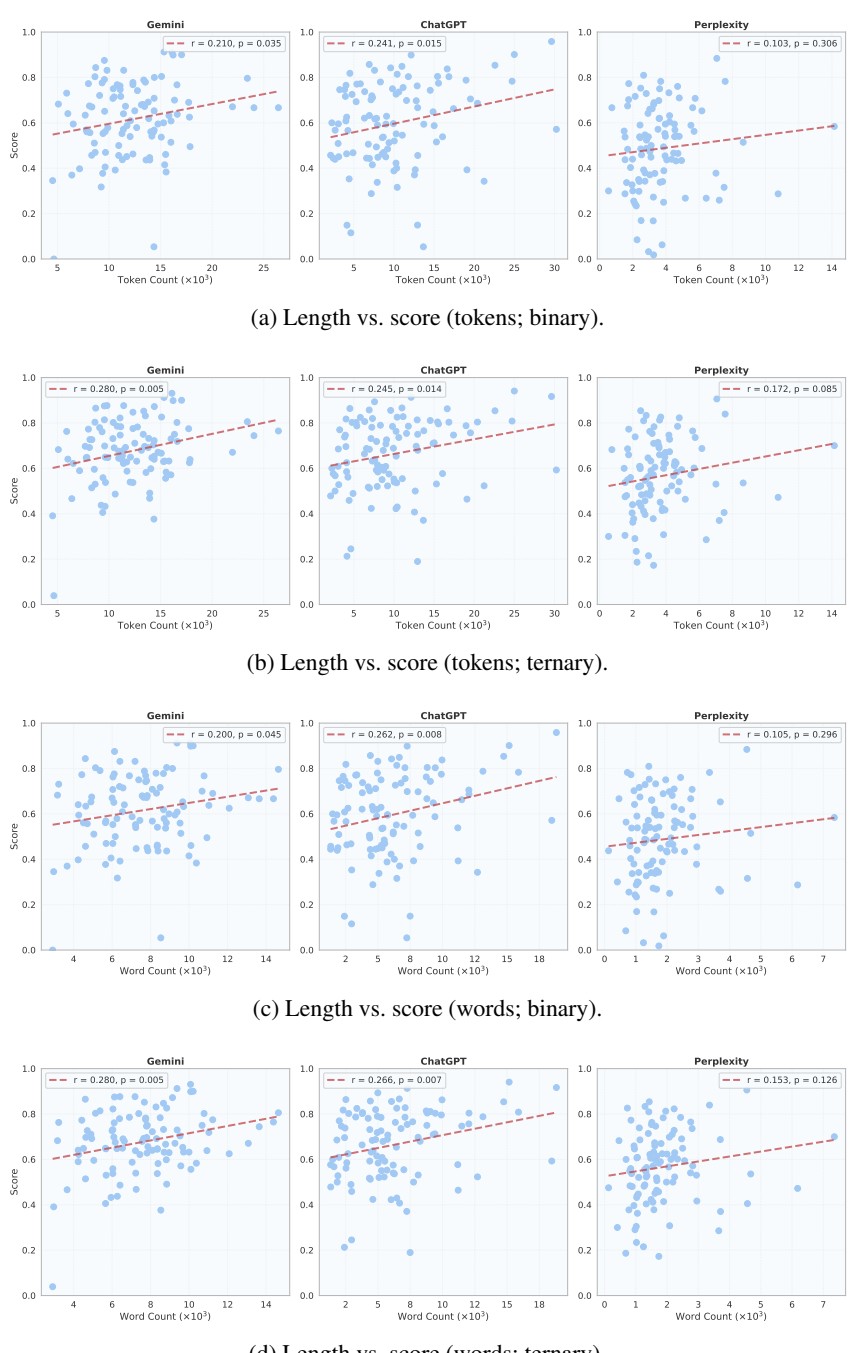

(a) Length vs. score (tokens; binary).

(b) Length vs. score (tokens; ternary).

(c) Length vs. score (words; binary).

(d) Length vs. score (words; ternary).

Figure 17: Comparison of length vs. score across token and word counts for binary and ternary settings.

(width). Tasks exceeding 4 sequential inference steps or 35 minutes of human-equivalent time show universal performance collapse across all evaluated systems (see Fig. 5). With RESEARCHRUBRICS averaging 25.7 criteria per prompt (see Fig. 9), approaching the $2^n - 1$ component complexity for $n = 5$ features, we operate near the theoretical saturation point for reliable evaluation.

**The Length-Quality Conflation Problem** Deep Research agents produce outputs 10-100 times longer than standard LLM responses (5,000-50,000+ tokens; see Table 9), raising questions about whether length drives perceived quality. Our criterion-level analysis reveals a nuanced relationship:

longer responses correlate with higher scores (see Fig. 17), but this primarily reflects legitimate information density rather than padding. Systems generating comprehensive reports with 100+ source synthesis necessarily require length, yet evaluators show documented bias toward verbosity independent of content quality.

RESEARCHRUBRICS' atomic evaluation partially mitigates this bias. Each of 2500+ criteria checks specific content presence rather than holistic impressions. However, the correlation persists even at the criterion level, suggesting that either (1) comprehensive responses naturally satisfy more criteria, or (2) length bias operates even on supposedly objective checkpoints. Distinguishing these explanations requires controlled experiments varying response length while holding information content constant.

**Architectural Limitations Beyond Prompt Engineering**   The consistency of failure patterns across systems—45-50% implicit criteria failures (see Fig. 4), poor multi-hop reasoning, synthesis bottlenecks—indicates fundamental architectural constraints rather than implementation differences. Multi-hop reasoning studies Yang et al. (2018) demonstrate that while agents achieve 80%+ success on first-hop inference, bridge entity resolution in early neural layers creates hard limits on subsequent reasoning depth. This explains the limited improvements from prompt engineering alone.

The breadth-accuracy trade-off further illustrates these constraints. No system successfully balances comprehensive coverage with precision. Gemini's 111-citation breadth sacrifices accuracy (81%) while Perplexity's 90% accuracy comes from restrictive 31-citation coverage. This isn't a tuning problem but reflects incompatible optimization objectives that current architectures cannot simultaneously satisfy.

## D   SUPPLEMENTARY FIGURES AND TABLES

We provide concise descriptions of the ten prompt domains used in RESEARCHRUBRICS in Table 10.

## E   PROMPTS

The prompt we sent to the LLM-as-a-judge can be found in Fig. 18.

We used two prompt templates in the ablation experiments: one for example removal and one for rubric augmentation. Both are shown below for reproducibility.

| Category | Description of Prompts |
|---|---|
| **AI & ML** | Tasks centered on artificial intelligence, machine learning, and data science, including model evaluation, algorithmic comparisons, ethical considerations, and emerging applications. Prompts often require synthesis of technical papers, applied case studies, and discussions of interpretability, safety, or deployment challenges in real-world AI systems. |
| **STEM** | Science, technology, engineering, and mathematics prompts outside core AI/ML domains. These tasks require synthesizing information from textbooks, research papers, or technical reports (e.g., explaining physical principles, analyzing chemical processes, or modeling engineering systems). |
| **General Consumer Research** | Everyday research with complex constraints (e.g., finding an apartment under budget, multi-factor product comparisons, travel itineraries, personal finance or legal advice, health-related questions requiring reputable sources). |
| **Technical Documentation** | Prompts involving explanation of complex technical concepts, code, or APIs using official documentation or repositories (e.g., troubleshooting a programming error with library docs, comparing software architecture patterns). |
| **Hypotheticals & Philosophy** | Open-ended prompts asking for speculation, hypotheticals, or philosophical analysis, often requiring synthesis of diverse viewpoints (e.g., *"How might society change if X. . . ?"*, ethical dilemmas, future predictions in technology). |
| **Historical Analysis** | Questions about historical events, figures, or periods that require pulling from archives, historical texts, and scholarly interpretations (e.g., analyzing causes of a historical conflict with primary source references). |
| **Business Planning & Research** | Prompts related to business or entrepreneurship (e.g., developing a go-to-market strategy, analyzing a company's financial health, legal considerations for a startup, HR or marketing plan), often requiring use of industry reports or case studies. |
| **Creative Writing** | Long-form creative tasks that incorporate factual elements or research (e.g., writing a historical fiction scene with accurate period details, or a sci-fi story grounded in real science). |
| **Current Events** | Prompts focused on recent or ongoing events, necessitating retrieval of up-to-date news or data (e.g., analysis of a recent policy change, comparison of current market trends). |
| **Other** | Miscellaneous prompts that do not neatly fit in the above categories, including cross-domain questions or niche topics. |

Table 10: Prompt domains in RESEARCHRUBRICS.

```
SYSTEM:
You are an expert evaluator tasked with assessing whether a document satisfies
specific rubric criteria.  Your evaluation must be precise, objective, and based
solely on the evidence present in the document.

## Evaluation Framework
You will evaluate each rubric criterion using a three-tier satisfaction scale:
1.  **Not Satisfied (Score:  0.0)**:  The document fails to meet the criterion.
Key elements are missing, incorrect, or inadequately addressed.
2.  **Partially Satisfied (Score:  0.5)**:  The document partially meets the
criterion.  Some elements are present but incomplete, lacking depth, or missing
important aspects.
3.  **Satisfied (Score:  1.0)**:  The document fully meets the criterion.  All
required elements are present, well-developed, and appropriately detailed.

## Evaluation Process
1.  **Understand the Criterion**:  Carefully read and interpret what the rubric
is asking for.
2.  **Search for Evidence**:  Systematically review the document for relevant
content that addresses the criterion.
3.  **Assess Completeness**:  Evaluate whether the evidence fully, partially, or
fails to satisfy the criterion.
4.  **Provide Reasoning**:  Explain your evaluation with specific references to
the document content.

## Important Guidelines
- Base your evaluation ONLY on what is explicitly present in the document
- Do not make assumptions about implied or missing content
- Consider the quality, completeness, and relevance of the evidence
- Be consistent in your evaluation standards across all criteria
- Provide specific examples from the document to support your verdict

Note:  Example lists in these rubrics are intended to illustrate possible
reasoning patterns or relevant topics.  These example lists contain correct
answers but are not exhaustive.  Use them as guidance, but also make your own
final judgment about what qualifies as correct when appropriate.

USER:
## Document Content
{document_content}

## Rubric Criterion to Evaluate
**Title**:  {rubric_title}
**Category**:  {rubric_category}
**Weight**:  {rubric_weigh}

## Your Task
Evaluate whether the above document satisfies this specific rubric criterion.

## Required Response Format
Provide your evaluation in the following JSON format:
```json
{
"verdict":  "[Not Satisfied/Partially Satisfied/Satisfied]",
"score":  [0.0/0.5/1.0],
"confidence":  [0.0-1.0],
"reasoning":  "Detailed explanation with specific evidence from the document",
"evidence_quotes":  ["Direct quote 1", "Direct quote 2", ...],
"missing_elements":  ["Element 1 that would improve satisfaction", ...]
```}

Ensure your response is ONLY the JSON object, with no additional text.
```

Figure 18: Prompt used for **example removal** during rubric preprocessing.

```
SYSTEM:
You are tasked with removing examples from a rubric text while
keeping everything else EXACTLY the same.

Your job is to:
1.  Identify portions of text that contain a list of examples,
typically in the form "(e.g., example1, example2, example3)" or
similar.
2.  Remove ONLY these example portions.
3.  Keep all other text, formatting, punctuation, and structure
EXACTLY the same.
4.  Do not rephrase, reword, or change anything else.
5.  Do not add any new content.
6.  Simply return the text with the example portions removed.

Examples of what to remove:
- "(e.g., a diagnosis code block, a free-text note snippet
without PHI, tabular data contexting text and numerical data)"
- "(i.e.  programmatic text extractions, more rigorous NLP and
machine learning techniques, etc.)"
- "((1) National Library of Medicine, (2) CDC Wonder or (3)
publications from well-known universities)"

Be very careful with maintaining the exact same structure and
wording for the rest of the rubric.

USER:
Please remove the examples from the following rubric text while
keeping everything else EXACTLY the same:

{rubric_text}
```

Figure 19: Prompt used for grading via the **LLM-as-judge** framework.

```
SYSTEM:
You are an expert at improving rubrics that are used to
evaluate model responses.  Make the rubrics more detailed, both
in terms of facts the models should cover and any definitions
or examples that should be added, while still keeping the
rubrics somewhat concise.

CRITICAL FORMATTING REQUIREMENTS:
- Return exactly ONE cohesive sentence (NO newlines, NO line
breaks).
- The rubric should be ONE SINGLE SENTENCE but can contain
multiple phrases, subparts, clauses, and run-on components.
- Target approximately 100 words on average, but you can exceed
that when necessary for completeness.
- Do NOT create multiline, paragraph-style, or bullet-point
rubrics.

IMPORTANT: You will receive exactly ONE rubric to improve,
and you must return exactly ONE enhanced version of that same
rubric.  Do not create multiple rubrics or variations.

Your job is to:
1.  Keep ALL original information from the rubric EXACTLY as it
is - do not delete or remove any core information, knowledge or
intent from the rubric.
2.  Make the rubric more detailed and concrete by adding
specific examples inline (e.g., specific answers or patterns
that might help the model to generalize)
3.  Clarify vague terms with more precise descriptions within
the same sentence flow.
4.  Add any information that may be missing.
5.  Make the rubric as actionable and unambiguous as possible
while staying concise.

Focus on adding inline:
- Concrete examples in parentheses (e.g., specific technical
details, data formats), which need not be exhaustive.
- Clear boundary conditions.
- Any definitions for unclear terms.

Do NOT:
- Remove any original content.
- Change the fundamental meaning or intent of any rubric.
- Add an entirely new rubric.
- Create multiple versions or variations (don't generate more
than one rubric output).
- Use newlines, bullet points, or multiline formatting.
- Break the rubric into multiple sentences.

Return only the single improved rubric as one cohesive
sentence.

USER:
Enhance this rubric by adding specific examples and details
while formatting it as ONE cohesive sentence (no newlines, but
the rubric can contain multiple phrases and clauses):

{rubric_text}

Return only the enhanced single-sentence rubric with no
additional text.
```

Figure 20: Prompt used for **LLM-based rubric augmentation**.

