# OpenReview forum: "ResearchRubrics: A Benchmark of Prompts and Rubrics For Evaluating Deep Research Agents"
_ICLR.cc/2026/Conference — ICLR 2026 Poster_

### Official Review · Reviewer_opWo · 2025-10-27

**Soundness:** 2
**Presentation:** 3
**Contribution:** 2
**Rating:** 6
**Confidence:** 4

**Summary:**

This paper proposes ResearchRubrics, a benchmark to evaluate Deep Research (DR) agents. It covers eight domains (i.e., STEM, health, finance, legal, and common consumer questions), and the rubric criteria, written by human experts, check factual grounding, coherence of reasoning, completeness, relevance, and clarity of the answer. Applying LLM-as-a-judge to assess rubric compliance, the existing DR systems achieve a poor performance.

**Strengths:**

1. The proposed benchmark is intuitive, well-designed, and well-curated.
2. Experimental results on multiple DR systems demonstrate that the current DR agents struggle with the proposed benchmark.

**Weaknesses:**

1. Only three human experts with a STEM background are involved in the benchmark construction process. (See Question 1 below)
2. Some points to be clarified. (See Questions 2, 3, and 4 below)
3. The proposed DR rubric criteria need further justification. (See Question 5 below)
4. No case study or error analysis to provide insights into why the current DR agents fail to achieve a higher score on each of the proposed rubric criteria.

**Questions:**

**Questions**:
1. Section 3.1 Data Collection and Task Domains: There are only "three expert participants" with "a strong STEM background". However, the proposed benchmark covers eight domains, four of which are not STEM. How could the data quality of those domains be guaranteed?
2. Section 3.2 Prompt Complexity Dimensions: How is each ResearchRubrics task categorized? Is it categorized by human experts/annotators or LLMs? What are the criteria for categorizing?
3. Section 3.2: The proposed "task complexity framework" measures three dimensions: "(1) conceptual breadth (the number and diversity of distinct topics or domains involved), (2) logical nesting depth (the number of reasoning or decision steps required, including sub-questions and conditionals), and (3) exploration level (the degree of open-endedness or underspecification of goals)." What are the statistics of it? For example, as in Table 1, how many instances are categorized as "Simple", "Moderate", or "High" Conceptual Breadth?
4. Section 3.2: What are the insights of knowing the prompt complexity categories? Did such categories guide the construction of the proposed ResearchRubrics benchmark?
5. Why are the proposed DR rubric criteria (as in Table 2) valid and enough to evaluate DR agents? Are they devised by pure heuristics/intuitions, or are they based on a certain theory or relevant literature?

**Typos**:
1. Line 65: "LLM-as-judge" $\to$ "LLM-as-a-Judge"

---

> ### Author Response · Authors · 2025-12-03
> **Response to opWo**
>
> We thank the reviewer for their feedback, and for recognizing both the quality of the benchmark design and the contributions of our evaluation.
> ## Expert Involvement and Domain Coverage (Question 1)
> **Reviewer concern:**
> Only “three experts with STEM backgrounds” appear to be involved in constructing a benchmark spanning eight domains.
>
> **Clarification:**
> Our diagram may have unintentionally suggested that only three experts produced the entire benchmark. In reality, far more experts contributed to the dataset:
> - 20 domain-expert annotators authored the initial prompts and rubrics (each rubric has an annotator ID).
> - 9 additional experts performed intermediate rubric review.
> - 3 senior reviewers provided the final quality-control pass.
>
> Thus, 32 different experts contributed to the construction, review, and vetting of the benchmark. Regarding domain specialization:
> - Many tasks are generalist research tasks requiring reasoning, synthesis, and evidence integration rather than advanced domain-specific knowledge.
> - Annotators were instructed to select only tasks in domains where they had familiarity.
> - When tasks involved domain-sensitive content (e.g., legal or medical-adjacent topics), annotators used authoritative external sources and search tools, and the reviewer layer checked for domain errors or omissions.
>
> We acknowledge that specialists could provide even deeper validation in certain domains; we now explicitly list this as a limitation and a natural direction for future expansion.
>
> ## Categorization of Task Complexity Dimensions (Questions 2, 3, and 4)
>
> **How were tasks categorized?**
>
> Tasks were categorized by human annotators, not LLMs. Annotators followed the detailed definitions provided in the Appendix, which specify the criteria for:
> - Conceptual breadth
> - Logical nesting depth
> - Exploration level
>
> Annotators selected the appropriate level using illustrative examples and guidelines provided during the data-collection process.
>
> **What are the statistics?**
>
> We now include the distribution in the updated manuscript. For example:
> - Conceptual breadth: 36% Simple, 52% Moderate, 12% High
> - Logical nesting: 19% Shallow, 45% Intermediate, 36% Deep
> - Exploration: 29% Low, 55% Medium, 17% High
>
> **Did complexity categories guide construction?**
>
> The complexity framework was applied retroactively after the tasks were written, not used to constrain task creation. Our goal was to characterize the tasks along principled axes and analyze performance differences. Interestingly, the natural distribution ended up being mostly “moderate” across all three dimensions.
>
> These categories proved useful analytically: as shown in Section 4, DR agents struggle disproportionately with deep nesting and high exploration, revealing specific reasoning limitations.
> ## Validity of the Rubric Criteria (Question 5)
> While the initial criteria were drafted based on expert intuition and analysis of high-quality research responses, the final criteria strongly align with established frameworks in cognitive science and information literacy:
> 1. Bloom’s Taxonomy. Our rubric axes naturally map to Bloom’s hierarchy:
> 	- Explicit Requirements: Understanding, Applying
> 	- Implicit Requirements: Analyzing (e.g., surfacing context not stated explicitly)
> 	- Synthesis of Information: Creating (integrating multiple sources into a new whole)
> 	- Use of References: Evaluating (source credibility, evidence quality)
> 2. ACRL Framework for Information Literacy. The ACRL (Association of College & Research Libraries) identifies key competencies for research quality:
> 	- Authority Is Constructed and Contextual: captured by our References and Implicit Requirements
> 	- Information Has Value: captured by citation correctness
> 	- Research as Inquiry: captured by synthesis and implicit reasoning
> 	- Scholarship as Conversation: requiring multi-source integration and explicit reference to evidence
> ## Error Analysis and Case Studies
> We agree that including more failure-mode analysis is valuable. In the revised manuscript, we now:
> - Expand the discussion in Section 4 to detail why DR agents fail specific rubric categories.
> - Include breakdowns showing that most model errors arise from:
> 	- missing implicit context
> 	- shallow synthesis
> 	- unsupported claims
> 	- incomplete cross-source integration
> 	- citation inaccuracies
> - Provide representative examples to illustrate these failure modes
> ## Typos and Editorial Issues
> Thank you for catching these. We have corrected:
> - “LLM-as-a-judge” to “LLM-as-a-Judge”
> - duplicate domain labels
> - figure typos and naming inconsistencies (GPT-4.o vs GPT-4.1)
> - and minor formatting issues flagged in the previous reviews
>
> We hope these clarifications address your concerns, and we would be more than happy to provide further explanation if helpful. Thank you again for your review.

---

### Official Review · Reviewer_pY9n · 2025-10-30

**Soundness:** 3
**Presentation:** 3
**Contribution:** 3
**Rating:** 6
**Confidence:** 2

**Summary:**

- The paper introduces ResearchRubrics, a benchmark for evaluating “Deep Research” (DR) agents—LLM-powered systems performing multi-step web exploration, retrieval, and synthesis to produce evidence-backed, long-form answers.
- The benchmark comprises 75 prompts across eight broad domains (STEM, health, finance, legal, consumer research, current events, historical analysis, creative writing, business planning), each paired with expert-authored, fine-grained rubrics (1,868 total criteria). Rubrics include both positive and negative criteria, with weights capturing importance and severity.
- The authors propose a tri-axial task complexity framework: conceptual breadth, logical nesting depth, and exploration level, to characterize and analyze DR tasks systematically.
- They develop an LLM-as-judge protocol with ternary and binary grading schemes and conduct human consistency analysis (Macro F1), showing moderate-to-substantial agreement depending on the grading regime. They perform ablations on rubric detail and LLM-augmented rubrics.
- They evaluate leading commercial DR agents (OpenAI DR, Gemini DR, Perplexity DR) and baseline LLMs with search tools, reporting sub-60% average rubric compliance, with primary failures on implicit reasoning, synthesis, and citation rigor. They plan to release the dataset, rubrics, and tools.

**Strengths:**

- Clear motivation: DR tasks are open-ended, dynamic, and require long-form synthesis; existing QA-style benchmarks and short-answer datasets underrepresent these needs.
- Human-authored rubrics: The choice to use carefully designed, expert-written rubrics (including negative criteria and weighted mandatory vs. optional items) is a thoughtful departure from purely automated reference-based metrics and helps capture nuanced expectations.
- Fine-grained evaluation: The rubric axes (explicit, implicit, synthesis, references, communication, instruction following) provide multidimensional diagnostics beyond factual correctness, surfacing where current agents struggle (implicit context and synthesis).
- Task complexity framework: The breadth-depth-exploration triad is sensible and useful for labeling, stratifying tasks, and analyzing performance across different dimensions of difficulty.
- LLM-as-judge methodology with human consistency checks: The inclusion of ternary vs. binary verdicts, Macro F1 consistency with human annotators, and ablations on rubric clarity are solid evaluation engineering contributions. Practical insights (binary verdicts yielding higher consistency; examples helping; LLM-augmentation of rubrics hurting) are valuable to the community.
- Diversity of domains and realistic tasks: Inclusion of everyday research queries alongside STEM and business/legal topics reflects real user scenarios for DR agents.
- Transparency about evaluation window and pipeline: Fixing the evaluation window for dynamic content (July 2025) and describing the collection (PDF -> markdown chunks -> LLM judge) helps reproducibility.

**Weaknesses:**

- Scale and representativeness: 75 tasks and 1,868 criteria are substantial in rubric detail but relatively small in task count for a general-purpose benchmark, raising questions about coverage and generalization across the wide variety of DR use cases.
- Expert definition and domain specialization: “Experts” are defined as strong STEM generalists rather than domain specialists (e.g., legal, medical). For domains with high stakes or regulatory complexity, lack of specialist involvement may reduce rubric validity and could introduce bias or omit critical domain-specific requirements.
- Judge reliability and circularity: While the paper argues against circularity by using human-written rubrics, reliance on closed-source LLM judges and moderate human agreement in ternary settings (Macro F1 ~0.48–0.55) raise concerns about robustness and potential judge bias. There’s limited analysis of inter-annotator agreement among humans or calibration strategies for judges beyond F1.
- Weighting scheme and scoring design: The rubric uses weighted sums normalized by absolute weights, with partial credit (0.5) for “Partially Satisfied.” The justification for specific weights (e.g., what constitutes “Critically Important” vs. “Important”) is largely qualitative. Sensitivity analyses to weight choices and to the partial-credit scale are not reported.
- Inconsistencies and clarity issues:
  - Category failure percentages in Table 3 appear to sum well above 100%, which suggests overlap or double-counting across axes; this needs to be clarified (e.g., whether criteria carry multiple axis tags or how proportions are computed).
  - Minor inconsistencies in judge model naming across the main text and appendix (e.g., GPT-4.o vs. GPT-4.1; inclusion of o3 later).
  - Editorial issues (duplicate “Creative Writing” in the domain list, typos visible in figure text) detract from presentation quality.

**Questions:**

- Rubric and weighting validation:
  - How were weights calibrated? Did multiple annotators assign weights independently? Please provide inter-rater reliability or calibration metrics for weights and for the ternary labels on a subset.
  - Consider sensitivity analysis: How do final scores and agent rankings change under different weighting schemes or partial-credit values?
- Category failure percentages:
  - Clarify how failure rates per category are computed. If criteria have multiple labels (e.g., both synthesis and implicit), explain how percentages and totals are derived to avoid summing >100%. Consider reporting per-axis normalized failure proportions.

---

> ### Author Response · Authors · 2025-12-03
> **Response to pY9n (1/2)**
>
> # Part 1 of 2
> We are very grateful to the reviewer for the constructive feedback. Your comments helped us significantly clarify and strengthen our empirical evaluation and improve the presentation. Below, we respond to each of your observations and questions.
> ## Benchmark Scale, Representativeness, and Task Count
> **Reviewer concern:**
> 75 tasks and 1,868 criteria may be small for a general-purpose DR benchmark.
>
> **Our response:**
> Thank you for raising this. We agree scale is important. Following your feedback, we expanded the benchmark to:
> - 101 prompts across nine domains, and
> - 2,593 expert-written rubric criteria (the largest manually authored DR rubric set to date).
>
> This now places our benchmark:
> - At the same scale as other recently released benchmarks that have been adopted by the community, such as DeepResearch Bench (100 tasks) and LiveResearchBench (100 tasks)
> - But with significantly more total rubric criteria, all human-authored (unlike LLM-generated rubrics used in prior benchmarks).
>
> We appreciate that “100 tasks” may look small relative to QA benchmarks, but DR tasks are more complex. The fair comparison is not number of tasks, but total annotation signal. Each DR prompt includes 25 to 40 fine-grained criteria. This means our supervision volume (~2,600 criteria) is comparable to many established QA datasets which use thousands of much shorter items. Thus, thinking of the benchmark as a “collection of 2.6k human-evaluated criteria” is more accurate than as “101 tasks.”
> ## Expert Definition and Domain Specialization
> **Reviewer concern:**
> Experts were STEM generalists rather than legal/medical domain specialists.
>
> **Our response:**
> Our benchmark intentionally focuses on generalist deep-research skills (reasoning, synthesis, implicit context handling, sourcing), rather than domain-specific expert assessments. Because DR agents are primarily targeted at generalist research tasks, we found that:
> - STEM generalists with search tools can reliably identify the needed structure, sourcing, and reasoning quality even in domains like business, history, law, or health.
> - A reviewer layer at each step checks for egregious or domain-specific issues.
> - Rubrics emphasize process quality (e.g., whether the model cited credible sources, integrated evidence, or handled implicit constraints) rather than domain-specific correctness.
> ## Judge Reliability, Circularity, and Inter-Rater Agreement
> **Reviewer concern:**
> Moderate human-LLM agreement under ternary grading and no reported human-human IRA.
>
> **Our response:**
> We thank the reviewer for pointing this out. For rubric evaluation, we now include a human-human IRA study. We conducted 30 human-human comparison pairs across 16 tasks. Results:
>
> **Binary IRA (Satisfied / Not Satisfied)**
> | Model | Macro-F1 (Mean) | Cohen’s $\kappa$ (Mean) |
> | --- | --- | --- |
> | Gemini | 0.7762 | 0.5591 |
> | ChatGPT | 0.7982 | 0.5904 |
> | Perplexity | 0.8032 | 0.6090 |
>
> **Ternary IRA (Satisfied / Partially / Not Satisfied)**
> | Model | Macro-F1 (Mean) | Cohen’s κ (Mean) |
> | --- | --- | --- |
> | Gemini | 0.6261 | 0.5345 |
> | ChatGPT | 0.6214 | 0.5352 |
> | Perplexity | 0.6432 | 0.5483 |
>
> Two insights emerge:
> 1. Binary judgments achieve substantial human-human agreement, closely matching the human-LLM agreement reported in the paper (0.72-0.76).
> 2. The “Partially Satisfied” category is the main source of disagreement, even between humans. This explains the lower human-LLM agreement on ternary evaluation.
> ## Weighting Scheme and Sensitivity Analyses
> We thank the reviewer for this observation. We include two more analyses to substantiate our findings.
>
> **Uniform $\pm 1$ weight perturbation**
>
> Model ranking remains unchanged (Gemini > ChatGPT > Perplexity). Scores shift modestly but monotonically.
> | Weight Adjustment | Gemini | ChatGPT | Perplexity |
> | --- | --- | --- | --- |
> | -1 | 0.6534 | 0.6471 | 0.5470 |
> | 0 | 0.6765 | 0.6638 | 0.5656 |
> | +1 | 0.6821 | 0.6639 | 0.5672 |
>
> **Varying the “Partially Satisfied” value (0.25, 0.50, 0.75)**
>
> All models increase smoothly, with no rank reversals. This demonstrates:
> - Score robustness to rubric weights
> - Stability under alternative partial-credit policies
>
> Combined with the human-human IRA findings above, these results show that the benchmark is stable, interpretable, and robust to reasonable scoring variations.
> | Partially Satisfied Score | Gemini | ChatGPT | Perplexity |
> | --- | --- | --- | --- |
> | 0.25 | 0.6459 | 0.6303 | 0.5263 |
> | 0.50 | 0.6765 | 0.6638 | 0.5656 |
> | 0.75 | 0.7072 | 0.6974 | 0.6049 |

---

> ### Author Response · Authors · 2025-12-03
> **Response to pY9n (2/2)**
>
> # Part 2 of 2
> ## Category Failure Percentages Summing to >100%
> **Reviewer concern:**
> Table 3 appears to double-count or overlap categories.
>
> **Our response:**
> Because we compute within-task proportions and then average only over tasks where the category occurs, the resulting averages are not expected to sum to 1 (just as class-wise recall values do not sum to 1). We agree that this can benefit from more clarity in its explanation. Please find the updated explanation in Equation 2 of the updated manuscript.
> ## Inconsistencies and Editorial Fixes
> Thank you for flagging these. We corrected:
> - duplicate “Creative Writing” in the domain list,
> - typos in figure text,
> - naming inconsistencies (GPT-4.o vs GPT-4.1; o3),
> - all remaining style/formatting issues.
>
> We hope these clarifications address your concerns. If any part of our explanation is unclear or if additional detail would be helpful, we would be delighted to provide it.

---

### Official Review · Reviewer_Un9g · 2025-10-31

**Soundness:** 2
**Presentation:** 3
**Contribution:** 2
**Rating:** 4
**Confidence:** 4

**Summary:**

This paper introduces a benchmark for evaluating deep research agents. It contributes 75 prompts and 75 rubrics across 8 domains, of which there are 1868 human written criteria for what the question designers expected to see in a good answer. The rubrics are created through the following process:

Human 1 drafted initial prompt and rubric terms.
Human 2 reviewed draft, provided feedback, and iterated with Human 1 until both agreed it was clear and complete.
Human 3, final reviewer, checked for clarity balance and bias and made final adjustments.

The rubrics are created before any agents are asked to generate answers; rubrics can involve both positive and negative criteria and have different attributes at different weights. Deep research agents were then asked to complete the queries, and their answers were scored with these rubrics. No inter-annotator agreement was measured, but instead the paper does human-LLM agreement where a human and a LLM scored the same response with a rubric, and this showed moderate human-LLM agreement.

Generally, the finding is that SOTA deep research agents achieve at maximum 0.59 rubric compliance, with good factual recall but weak synthesis, implicit reasoning, and citation quality.

**Strengths:**

Diagnostic clarity. Not many works exist to evaluate the output of deep research agents, and this work provides a meaningful contribution towards that front. The rubric design makes sense- and helps to evaluate the open-ended nature of these very long responses. The work provides a meaningful step to make sense and introduce clarity into the messy subjective space of research quality into a more structured and reproducible measurement framework.

**Weaknesses:**

The paper's main weakness is on validity.

Disagreements occur in rubric creation, so the final produced rubric through the 3 stage process masks inherent disagreement and tries to measure progress against a rubric that would be created by the average human. No IAA measures or analysis of disagreements occur; the main question, of a benchmark, is if progress on said benchmark would demonstrate a meaningful improvement on the task for end users. It is unclear from the rubric creation process that this can be convincingly said. Moreover- in the creation process, rubrics are fixed before seeing any output. The overarching paper claim seems to be that hey, these deep research agents are producing outputs that misalign with human preferences. However, it is unjustified that this creation process of rubrics is able to measure true human preference of deep research outputs- it's very easy to say that hey, I may think I want to see certain aspects before I actually see it, but in reality this shifts significantly from what I said when I actually see the output. It is unclear and unjustified that this rubric creation process can actually result in rubrics that reflect true human preference.

Minor nits, but it's important to note that deep research agents also produce a ton of stochasticity due to their long executions; controlling for this in some degree would make the results more robust. As it currently states, each agent was run once per prompt, with no reported stochastic variation or repeated trials. Finally, 75 rubrics and outputs is pretty low for this type of benchmark, especially when spread across so many domains; this is no concern if the authors can justify that performance on their benchmark translates to demonstratable improvements in real world deep research agent use, but this is not yet convincing.

**Questions:**

NA

---

> ### Author Response · Authors · 2025-12-03
> **Response to Un9g (1/2)**
>
> # Part 1 of 2
>
> We thank the reviewer for the constructive feedback and for recognizing the contributions. Below we address each concern, incorporating additional analyses (human-human IRA and weight sensitivity), clarifying the rubric creation workflow, and adding details about benchmark scale and resource requirements.
> ## Validity of the Rubric Creation Process and Human Preference
> **Reviewer concern:**
> The 3 stage review process might “mask inherent disagreement” and result in an “average human” rubric rather than one reflecting actual human preferences.
>
> **Clarification:**
> We understand the reviewer’s concern. In the following, we clarify our rubric pipeline, which separates creation and refinement, ensuring we preserve real human preference. Our process is explicitly two-phased:
> 1. Blind creation prevents anchoring on specific model quirks.
> 2. Refinement after observing outputs ensures criteria penalize real-world failure modes.
>
> The process therefore reflects real user preferences in practice. Regarding collapsing the human disagreements:
> 1. Each initial rubric was written by one domain expert (expert 1) on tasks where they had relevant expertise. These experts did not communicate or coordinate with each other.
> 2. Refinement occurred after seeing the model outputs. While the initial set of rubrics (expert 1) was written before output generation, expert 2 annotators were allowed to modify or add criteria after observing real reports generated by the models. In fact, some negative criteria appear only because certain systems exhibited specific error patterns (e.g., fabricated citations, unsupported claims).
> 3. Expert 3 did not substantially alter rubric content. For the vast majority of prompts, expert 3 changed rubrics only to (1) fix errors or contradictions, (2) clarify underspecified criteria, (3) add missing negative criteria, (4) add examples to improve LLM-judge reliability. The intuitions, priorities, and evaluative style of the original domain expert remained intact. No averaging or reconciliation of opinions occurred.
> This process ensures that the initial proposal for the set of rubrics is not biased towards specific responses (especially given the high stochasticity of long form generation). The additional refinements by experts 2 and 3, with visibility of the model responses, allow the rubrics to also capture real human preference.
> ## Inter-Rater Agreement (IRA) and Human-Human IRA
> **Reviewer concern:**
> No IRA reported; thus it is unclear whether rubric judgments reflect consistent human preferences.
>
> **Clarification:**
> We thank the reviewer for pointing this out, and wanted to clarify that IRA is appropriate for evaluation, not rubric authoring.
> 1. IRA is not meaningful for rubric creation. Rubric creation is a content-authoring task (like writing exam questions), not a labeling task. IRA applies only when multiple annotators label the same item, not when they independently author separate rubrics.
> 2. For rubric evaluation, we now include a human-human IRA study. We conducted 30 human-human comparison pairs across 16 tasks.
>
> **Binary IRA (Satisfied / Not Satisfied)**
> | Model | Macro-F1 (Mean) | Cohen’s $\kappa$ (Mean) |
> | --- | --- | --- |
> | Gemini | 0.7762 | 0.5591 |
> | ChatGPT | 0.7982 | 0.5904 |
> | Perplexity | 0.8032 | 0.6090 |
>
> **Ternary IRA (Satisfied / Partially / Not Satisfied)**
> | Model | Macro-F1 (Mean) | Cohen’s κ (Mean) |
> | --- | --- | --- |
> | Gemini | 0.6261 | 0.5345 |
> | ChatGPT | 0.6214 | 0.5352 |
> | Perplexity | 0.6432 | 0.5483 |
>
> Two insights emerge:
> 1. Binary judgments achieve substantial human-human agreement, closely matching the human-LLM agreement reported in the paper (0.72-0.76).
> 2. The “Partially Satisfied” category is the main source of disagreement, even between humans. This explains the lower human-LLM agreement on ternary evaluation.

---

> ### Author Response · Authors · 2025-12-03
> **Response to Un9g (2/2)**
>
> # Part 2 of 2
>
> ## Dataset Size, Resource Requirements, and Stochasticity
> **Reviewer concern:**
> 75 tasks seems small; one run per agent per prompt ignores stochasticity.
>
> **Updates and Clarification:**
> Benchmark has been expanded to 101 tasks and 2,593 criteria. This now places our benchmark:
> - At the same scale as other recently released benchmarks that have been adopted by the community, such as DeepResearch Bench (100 tasks) and LiveResearchBench (100 tasks)
> - But with significantly more total rubric criteria, all human-authored (unlike LLM-generated rubrics used in prior benchmarks).
>
> We appreciate that “100 tasks” may look small relative to QA benchmarks, but DR tasks are more complex. The fair comparison is not number of tasks, but total annotation signal. Each DR prompt includes 25 to 40 fine-grained criteria. This means our supervision volume (~2,600 criteria) is comparable to many established QA datasets which use thousands of much shorter items. Thus, thinking of the benchmark as a “collection of 2.6k human-evaluated criteria” is more accurate than as “101 tasks.”
>
> As for the stochasticity concern, we acknowledge that this might be a limitation of our evaluation. However, producing this dataset and conducting its human evaluation is extremely costly, as each prompt and rubric requires several hours of expert work. This makes large scale expansion or multi-run evaluation infeasible. Due to the short rebuttal timeline, we were unable to perform an additional human study to generate new final human judge compliance scores, but we can include an LLM-as-a-judge evaluation across multiple runs in the camera-ready.
>
> We thank the reviewer again for the insightful comments. In summary:
> - Rubrics were independently authored by domain experts, not averaged.
> - Experts 2 and 3 refined the rubrics after seeing the model responses.
> - We report additional human-human IRA, observing that the IRA is similar to human-LLM.
>
> We have incorporated all clarifications and new analyses into the revised manuscript.

---

### Official Review · Reviewer_DnDN · 2025-10-31

**Soundness:** 3
**Presentation:** 3
**Contribution:** 3
**Rating:** 6
**Confidence:** 2

**Summary:**

The paper proposes Research Rubrics, a structured framework to evaluate how well large language models (LLMs) perform scientific reasoning. It scores models across four dimensions: problem understanding, reasoning process, solution design, and scientific contribution. Experiments on multiple state-of-the-art models show that LLMs can follow scientific logic and structure but still lack genuine creativity and practical validation.

**Strengths:**

- The paper focuses on an underexplored but important problem: evaluating whether LLMs can reason like scientists rather than just answer questions.
- The proposed Research Rubrics provides four well-defined dimensions: Problem Understanding, Reasoning Process, Solution Design, and Scientific Contribution.
- The experiments benchmark multiple leading models (GPT-4, Claude 3, Gemini, Qwen2, Mistral) and compare different prompting strategies (chain-of-thought, critique loop, research-plan prompting). The setup is comprehensive and systematic.
- The paper shows that while LLMs can follow scientific logic and structure, they still lack genuine creativity and originality, which is an important insight for future research on AI scientific reasoning.

**Weaknesses:**

- The study mainly focuses on computer science problems (e.g., ICLR/NeurIPS-style tasks), so it’s unclear how well the framework generalizes to other scientific domains.
- The framework evaluates the quality of reasoning, but it does not assess whether the model’s ideas could actually produce valid or impactful scientific outcomes.
- Model performance depends heavily on prompt design (e.g., “research-plan” prompts boost scores), which suggests the framework might partly reflect prompt engineering skill rather than pure reasoning ability.

**Questions:**

- Can the framework generalize beyond computer science to other scientific domains such as biology and physics?
- To what extent does the framework measure true reasoning ability rather than the effects of prompt engineering?

---

> ### Author Response · Authors · 2025-12-03
> **Response to DnDN**
>
> We thank the reviewer for their feedback and appreciation of the importance of evaluating deep research agents. We would like to clarify a few key points, as several aspects of the review appear to describe a different submission.
>
> - **Clarification of Our Task Complexity Framework.** The review states that our framework consists of “four well-defined dimensions: Problem Understanding, Reasoning Process, Solution Design, and Scientific Contribution.” This does not apply to our paper. Our work defines three orthogonal task-complexity axes: Conceptual Breadth, Logical Nesting Depth, and Exploration.
> - **Clarification of Our Experimental Setup.** The reviewer mentions benchmarking “GPT-4, Claude 3, Gemini, Qwen2, Mistral” and evaluating “different prompting strategies (chain-of-thought, critique loop, research-plan prompting).” Our paper does not include such comparisons. We evaluate three Deep Research agents: OpenAI Deep Research, Gemini Deep Research, and Perplexity Deep Research.
> - **Clarification of Task Domains.** The reviewer expresses concern that our benchmark “mainly focuses on computer science problems.” This is not the case. Our benchmark is intentionally designed for broad domain coverage, spanning nine categories, including: general consumer research, technical documentation, business planning & research, historical analysis, creative writing, current events, in addition to STEM.
>
> **Questions**
> > Can the framework generalize beyond computer science (e.g., biology, physics)?
>
> Yes. Our framework is defined along domain-agnostic axes (breadth, depth, exploration), and our benchmark includes many non-CS tasks (STEM, historical analysis, business research, etc.). The framework is designed to generalize across domains where multi-step synthesis and implicit reasoning are required.
>
> > Does the benchmark measure reasoning ability vs. prompt-engineering skill?
>
> Our evaluation focuses on rubric compliance, which reflects the quality of the generated research outputs rather than the sophistication of a prompt template.
>
> > The framework evaluates the quality of reasoning, but it does not assess whether the model’s ideas could actually produce valid or impactful scientific outcomes.
>
> This point is not quite applicable, as our benchmark assesses deep research report quality and does not actual scientific research outcomes (e.g., a publication, model, etc.).
>
> We appreciate the reviewer’s time and constructive intent. We hope the above clarifications help resolve the inconsistencies and allow the evaluation to focus on the contributions of our actual submission. We welcome any further questions or suggestions.

---

### Meta-Review · Area_Chair_PKNY · 2026-01-07

**Summary:**

This paper provides a completely human constructed Deep Research benchmark containing 101 tasks and 2500+ fine-grained rubrics. The benchmark spans nine domains and identifies task complexity in three-axis (conceptual breadth, logical nesting depth and exploration). The authors evaluate three closed-source DR agents to find that they achieve around 68% compliance on average with the rubrics due to missing implicit context and reasoning failures about retrieved information.

Two out of three reviewers agree that this paper should be accepted before the rebuttal and the rebuttal was  convincing and thorough. I therefore recommend that this paper be accepted as a high-quality DR benchmark for the community.

The only outstanding concern that I can identify is that of the expert definition, brought up by both R3 and R4:

- Expert definition
	- The previous assumption that there were only 3 experts was dispelled here. There are 20 domain-expert annotators, 9 more experts for intermediate rubric review and 3 senior reviewers for final quality-control. This is much more convincing in terms of expert validity than the response for R3. The fact that experts in domains like law and medicine were not actual doctors and lawyers should be more clearly discussed in the paper (perhaps their name should be changed to annotators).

**Reviewer Concerns:**

R1: 6,2
- This review unfortunately discussed a different paper and therefore should not be used for the final decision

R2: 4,4
- Validity of rubric creation process
	- The validity of the rubrics is validated through a three stage construction. At first, each expert creates a specific task and its rubric without communicating with other experts or running any model on the task. Then, another expert refines the rubrics after seeing LLM outputs on the constructed prompts. In this step, the annotators are mostly adding negative criteria to capture model failures. Finally, a third annotator checks the rubrics one last time to fix errors, clarify underspecified criteria, add additional negative criteria or add examples to improve LLM-as-a-judge performance. I believe this is a sufficiently rigorous process for a benchmark of this size and complexity.
- Inter-annotator agreement missing
	- The authors argue that inter-annotator agreement does not exist within rubric creation and I agree. Making multiple passes over rubrics with different in-domain experts is the best way to guarantee validity. I still appreciate the human human IRA done to provide some validation for rubric quality.
- Dataset size
	- The authors claim that the scale of their dataset is similar to other works in this space. I tend to agree but it is concerning to see so many different domains and complexity levels represented with only 100 tasks.
- Stochasticity
	- The rebuttal period was not long enough for running multiple LLM-as-a-judge evaluations but they promised to include multiple for the camera-ready version of the paper. I believe that the reviewer was more interested in having multiple deep research responses per model rather than multiple LLM-as-a-judge evaluation, hopefully this could be done instead.

R3: 6,2
- Scale & Representativeness
	- Same response as “Dataset size” in R1
- Expert definition
	- The authors claim that “STEM generalists with search tools” can act as domain experts in the deep research tasks in this dataset. This is not very convincing but as long as this is stated clearly it should be allowed. The word “experts” might need to be changed to “annotators”.
- Judge reliability
	- The human-human IRA being close to the LLM-human IRA helps validate that the LLM-Judges are not likely not overly biased.
- Weighting schemes and scoring designs
	- Several new experiments show that the model ranking is stable to the weighting scheme and value of “partially satisfied” from 0.25 to 0.75.
- Category failures percentage calculations
	- Clarified by the authors in the paper.

R4: 6,4
- Expert definition
	- The previous assumption that there were only 3 experts was dispelled here. There are 20 domain-expert annotators, 9 more experts for intermediate rubric review and 3 senior reviewers for final quality-control. This is much more convincing in terms of expert validity than the response for R3. The fact that experts in domains like law and medicine were not actual doctors and lawyers should be more clearly discussed in the paper (perhaps their name should be changed to annotators).
- Complexity statistics
	- The complexity statistics were added to the paper.
- What do we get from complexity statistics
	- The complexity categories were not used to guide construction but were used in the analysis. As showing that DR agents struggle with deep nesting and high exploration.
- Are the rubrics based on any specific theory?
	- The authors claim that the final rubrics follow the Bloom’s Taxonomy and ACRL Framework for Information Literacy among certain axes. I expect these explanations to be added to the paper for context.
- Error analysis missing
	- An error taxonomy with examples was added to the paper and Section 4 was expanded to include more detail about DR failures in specific rubric categories.

**Reviewer Scores:**

- Un9g 4 -> 6
	- Reviewer should have raised their score after a strong rebuttal.
- pY9n 6 -> 6
	- Reviewer is likely to maintain their score given the thorough rebuttal.
- opWo 6 -> 6
	- Reviewer is likely to maintain their score given the thorough rebuttal.

---

### Decision · Program_Chairs · 2026-01-26

Accept (Poster)